# Nucleotide mismatches prevent intrinsic self-silencing of hpRNA transgenes to enhance RNAi stability in plants

Daai Zhang[1,2], Chengcheng Zhong[1], Neil A. Smith[1], Robert de Feyter[1], Ian K. Greaves[1], Steve M. Swain [1], Ren Zhang[2] & Ming-Bo Wang [1✉]

Hairpin RNA (hpRNA) transgenes are the most successful RNA interference (RNAi) method in plants. Here, we show that hpRNA transgenes are invariably methylated in the inverted-repeat (IR) DNA and the adjacent promoter, causing transcriptional self-silencing. Nucleotide substitutions in the sense sequence, disrupting the IR structure, prevent the intrinsic DNA methylation resulting in more uniform and persistent RNAi. Substituting all cytosine with thymine nucleotides, in a G:U hpRNA design, prevents self-silencing but still allows for the formation of hpRNA due to G:U wobble base-pairing. The G:U design induces effective RNAi in 90–96% of transgenic lines, compared to 57–65% for the traditional hpRNA design. While a traditional hpRNA transgene shows increasing self-silencing from cotyledons to true leaves, its G:U counterpart avoids this and induce RNAi throughout plant growth. Furthermore, siRNAs from G:U and traditional hpRNA show different characteristics and appear to function via different pathways to induce target DNA methylation.

[1] CSIRO Agriculture and Food, Clunies Ross Street, Canberra ACT 2610, Australia. [2] School of Chemistry & Molecular Bioscience, University of Wollongong, Wollongong, NSW 2522, Australia. ✉email: ming-bo.wang@csiro.au

RNA silencing is an evolutionarily conserved gene silencing mechanism in eukaryotes, where long dsRNA is processed by dicer or dicer-like (DCL) proteins into 20–30 nucleotide (nt) small RNA (sRNA) that induces RNA degradation via sequence complementarity[1–3]. In plants, multiple RNA silencing pathways exist, including microRNA (miRNA), trans-acting small interfering RNA (tasiRNA), repeat-associated siRNA (rasiRNA), and exogenic (virus and transgene) siRNA (exo-siRNA) pathways[4]. miRNAs are 20–24 nt sRNAs processed in the nucleus by DCL1 from short self-folding RNAs transcribed from *MIR* genes[2]. tasiRNAs are 21 nt secondary siRNAs derived from DCL4 processing of dsRNA synthesized by RNA-dependent RNA polymerase 6 (RDR6) from miRNA-cleaved *TAS* RNA fragment[4]. The 24-nt rasiRNAs are generated from repetitive DNA in the genome by the combined function of DNA-dependent RNA polymerase IV (Pol IV), RDR2, and DCL3[5]. The exosiRNA pathway overlaps with the tasiRNA and rasiRNA pathways and both DCL4 and DCL3 are involved in exosiRNA processing. In addition to DCL1, DCL3 and DCL4, plant genomes encode DCL2 or equivalent, which generates 22-nt siRNAs including 22-nt exosiRNAs, and plays a key role in systemic and transitive gene silencing in plants[6]. All of these plant sRNAs are methylated at the 2′-hydroxyl group of the 3′ terminal nucleotide by HUA Enhancer 1 (HEN1), which is thought to stabilize the sRNAs[7]. miRNAs, tasiRNAs and exosiRNAs are functionally similar to sRNAs in animals, and involved in post-transcriptional RNA degradation. rasiRNAs, however, function to direct de novo cytosine methylation at the cognate DNA, a transcriptional gene silencing mechanism known as RNA-directed DNA methylation (RdDM)[5]. The post-transcriptional RNA silencing mechanism has been extensively exploited as a gene knockdown technology in various eukaryotic systems, generally referred to as gene silencing or RNA interference (RNAi) technologies. In plants, the different RNA silencing pathways have led to different technical approaches, such as artificial miRNA, artificial tasiRNA, and virus-induced gene silencing technologies[4]. Long hpRNA transgenes, designed to express long hairpin-structured dsRNA, are the most widely used RNAi technology in plants, and a variety of successful applications of this technology have been demonstrated in plant biotechnology[4]. It can be anticipated that this RNAi approach will continue to be a powerful tool in many areas of crop improvements such as host-induced RNAi against pests and pathogens and metabolic engineering of novel traits through spatial and temporal gene knockdown, which is difficult to achieve using gene knockout technologies such as the CRISPR/Cas9 approach.

An hpRNA construct typically consists of a perfect inverted-repeat (IR) of a target gene sequence (forming the dsRNA stem of hpRNA) separated by a spacer sequence (forming the loop). Tandem DNA repeats, particularly the IR DNA structures, are widely observed to attract strong DNA methylation causing transcriptional silencing[8,9]. Beside the IR structure, siRNAs derived from hpRNA transgenes can potentially direct DNA methylation to their own sequence via the RdDM pathway[10,11]. hpRNA transgenes, therefore, differ from normal transgenes and are potentially subject to self-induced transcriptional silencing. Indeed, a previous study showed that hpRNA transgene-induced RNAi in Arabidopsis was enhanced in an RdDM mutant and that this enhanced RNAi effect correlated with reduced DNA methylation spanning from the IR DNA to the upstream promoter sequence[12]. An RNAi design that can prevent self-induced silencing would therefore be desirable for achieving durable and potent RNAi in plants.

In this study, we show that introducing nucleotide mismatches to disrupt IR DNA structure results in uniform and persistent RNAi in plants. Our results indicate that the traditional hpRNA transgenes with a perfect IR structure are generally prone to self-induced methylation and transcriptional silencing (referred to as self-silencing hereafter) causing large variability in RNAi efficacy, whereas the enhanced RNAi effect of mismatched hpRNA constructs is due to the prevention of methylation in both the IR and the promoters preventing self-silencing. In addition, we provide evidence that the IR-associated DNA methylation and self-silencing is not affected in two mutants of the RdDM pathway, and that siRNAs from G:U hpRNA transgenes are processed and function differently from traditional hpRNA transgenes, providing insights into IR-induced gene silencing in plants.

## Results

**Evenly mismatched and G:U base-paired hpRNA induce uniform RNAi.** We first tested three mismatched constructs in *Nicotiana tabacum* lines PPGH11 and PPGH24 expressing the β-glucuronidase (*GUS*) reporter gene as the RNAi target (Fig. 1a). These constructs contained the same 200 bp antisense wild-type (WT) *GUS* sequence as the traditional hpRNA construct (hpGUS[WT]) to ensure perfect sequence complementarity between antisense siRNAs and target *GUS* mRNA. The mismatched construct hpGUS[1:4] had one nucleotide substitution in every 4 nucleotides of the 200 bp sense sequence; hpGUS[2:10] contained 2 consecutive nucleotide substitutions in every 10 nucleotides; and hpGUS[G:U] had all 52 cytosine (C) nucleotides changed to thymine (T) nucleotides (Fig. 1a; Supplementary Fig. 1). The C to T changes in hpGUS[G:U] disrupted the perfect IR DNA structure but did not prevent the formation of perfect hpRNA due to G:U wobble base-pairing.

The hpGUS[WT] transgenic population showed a wide range of RNAi efficiency, with 35 of the 59 independent lines analyzed (59.3%) showing strong RNAi (GUS activity ≤10% of the PPGH11 and PPGH24 GUS-expressing control lines), 9 showing weak RNAi (GUS activity 10–30% of the control), and 15 almost no silencing (Table 1 and Fig. 1b), which was typical for traditional hpRNA constructs[13]. The hpGUS[2:10] construct behaved more like hpGUS[WT], inducing strong GUS RNAi in some lines (28 of 41, or 68.3%) but giving almost no GUS RNAi in the remaining 13 plants.

In clear contrast, 72 of the 74 hpGUS[G:U] lines (95.9%) tested showed strong RNAi (Fig. 1b). This uniform RNAi was not due to a uniform transgene insertion pattern across the independent lines: 16 randomly selected *GUS*-silenced lines showed a wide range of transgene copy numbers (Supplementary Fig. 2). All 33 hpGUS[1:4] lines showed *GUS* RNAi, although only 10 (30.3%) showed strong RNAi. Thus, relatively evenly distributed nucleotide substitutions, as in hpGUS[G:U] and hpGUS[1:4], allowed for uniform RNAi across transgenic populations. The relatively weak RNAi effect by hpGUS[1:4] coincided with its dsRNA stem having the lowest predicted thermodynamic stability (Fig. 1c). Consistently, there appeared to be a good correlation between the extent of *GUS* RNAi and the predicted dsRNA stability of the four hpRNAs (Fig. 1c).

As the G:U hpRNA construct induced strong and uniform RNAi against *GUS*, we tested this design against two endogenous genes in Arabidopsis, *ETHYLENE INSENSITIVE* 2 (*EIN2*) and *PHYTOENE DESATURASE* (*PDS*), silencing of which can be scored based on hypocotyl length of dark-germinated seedlings on 1-aminocyclopropane-1-carboxylic acid (ACC) medium[14] and photo-bleaching[15], respectively. The hpRNA constructs (Figs. 2a and 3a) were designed to target a 200 bp *EIN2* and 450 bp *PDS* mRNA sequences that contained 43 and 82 cytosines, creating 21.5% and 18.2% G:U wobble base-pairs for hpEIN2[G:U] and hpPDS[G:U], respectively (Supplementary Fig. 3).

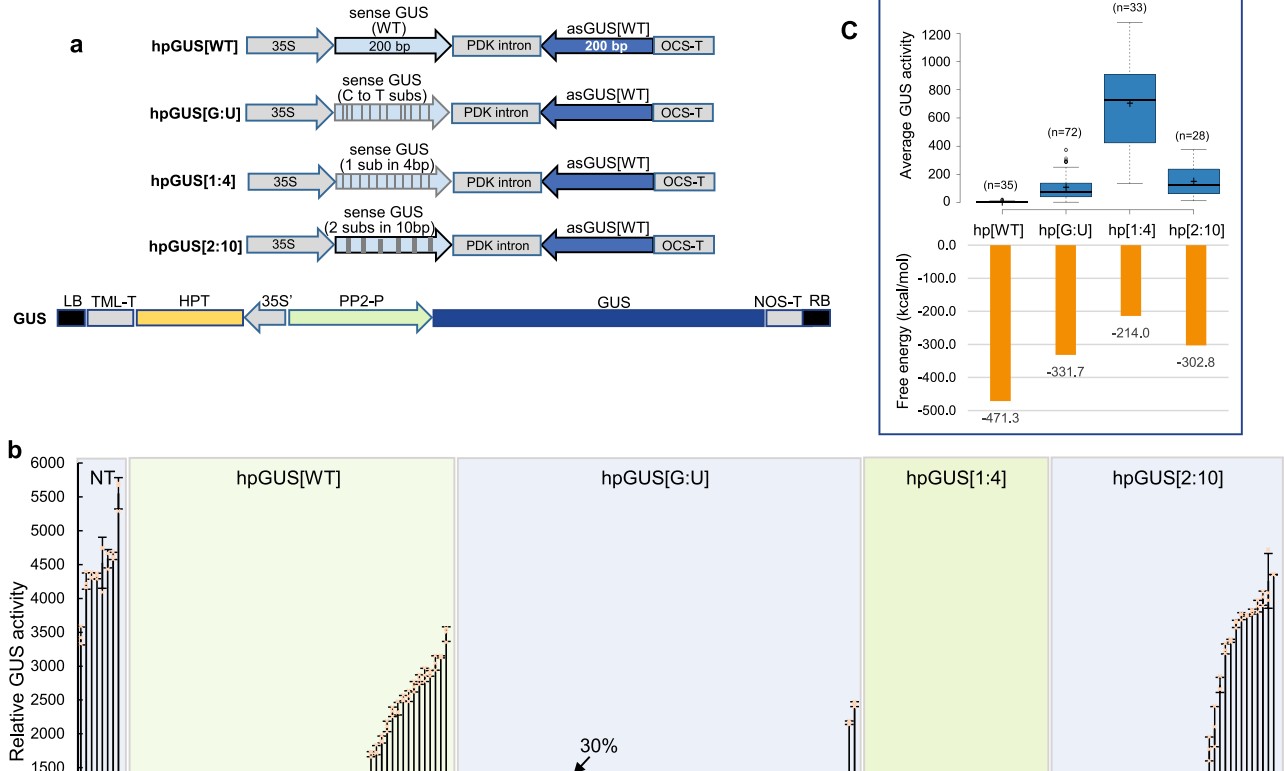

**Fig. 1 G:U and 1:4 mismatched GUS hpRNA constructs induce uniform GUS RNAi in tobacco. a** Schematic diagram of the four GUS hpRNA constructs and the target GUS gene in PPGH11 and PPGH24 plants. 35S, the longer version (~1.3 kb) of the cauliflower mosaic virus 35S promoter; 35S', a shorter version (337 bp from nt. −285 to +50) of the 35S promoter from pTRA151 vector; OCS-T, NOS-T, and TML-T, the transcriptional terminator sequences of the *Agrobacterium tumefaciens* octopine synthase, nopaline synthase, and tumor morphology large genes, respectively. HPT, hygromycin phosphotransferase gene; PP2-P, the *Cucurbit pepo* PP2 protein gene promoter. **b** GUS activity in independent T0 plants transformed with the conventional and modified hpRNA constructs. Each bar represents an independent T0 transgenic plant. Three technical replicates were measured for each line, and data are presented as mean values ± s.d. with the three data points shown as orange cross. The dashed green and pink lines indicate the 30 and 10% GUS activity levels of the untransformed PPGH11 and PPGH24 plants (NT). **c** The average levels of *GUS* silencing in the strongly silenced lines of hpGUS[WT], hpGUS[G:U] and hpGUS[2:10] and all the silenced plants of hpGUS[1:4] show a good correlation with the thermodynamic stability of predicted hpRNA structures. The boxplots (BoxPlotR, http://shiny.chemgrid.org/boxplotr/) above show the distribution of GUS activities of 35 hpGUS[WT], 72 hpGUS[G:U], 33 hpGUS[1:4], and 28 hpGUS[2:10] lines. The central horizontal line indicates the median value; the lower and upper borders of the box represent the first and third quartiles; whiskers extend 1.5 times the interquartile range from the first and third quartiles; outliers are represented by dots; crosses represent sample means. Source data are provided as a Source Data file.

**Table 1 Summary of *GUS* gene silencing by the four hpRNA constructs based on MUG assay data.**

| Constructs | Total No. transgenic lines | Strong silencing (<10% of control) | Weak silencing (10–30% of control) | Almost no silencing (>30% of control) |
|---|---|---|---|---|
| hpGUS[WT] | 59 | 35 (59.3%) | 9 (15.3%) | 15 (25.4%) |
| hpGUS[G:U] | 74 | 71 (95.9%) | 1 (1.4%) | 2 (2.7%) |
| hpGUS[1:4] | 33 | 10 (30.3%) | 23 (69.7%) | 0 |
| hpGUS[2:10] | 41 | 28 (68.3%) | 0 | 13 (31.7%) |

Analysis of 20 randomly selected hpEIN2 lines (100–200 T2 progeny for each line) showed that the hpEIN2[WT] lines had a high range of *EIN2* RNAi levels, with 7 lines (# 2, 5, 9, 10, 12, 14, and 16) showing low levels of RNAi (short hypocotyl length), and the other 13 lines (65%) having moderate to strong *EIN2* RNAi (Fig. 2b–d). In contrast, 18 of the 20 hpEIN2[G:U] lines (90%) displayed relatively uniform and strong *EIN2* RNAi. Individual siblings within each of the 18 lines also appeared to show less

variation in hypocotyl length than those of the hpEIN2[WT] lines (Fig. 2b–d), suggesting a greater sibling uniformity of RNAi.

Like the hpGUS[G:U] lines, the uniform *EIN2* RNAi in the hpEIN2[G:U] lines was in general not dependent on the number of transgene insertion (as judged by the kanamycin resistant:-sensitive ratio of the T2 progeny plants). The 18 hpEIN2[G:U] lines with strong RNAi had a range of insertion numbers, including high-copy number insertions (5:1 to 86:1 ratios), with

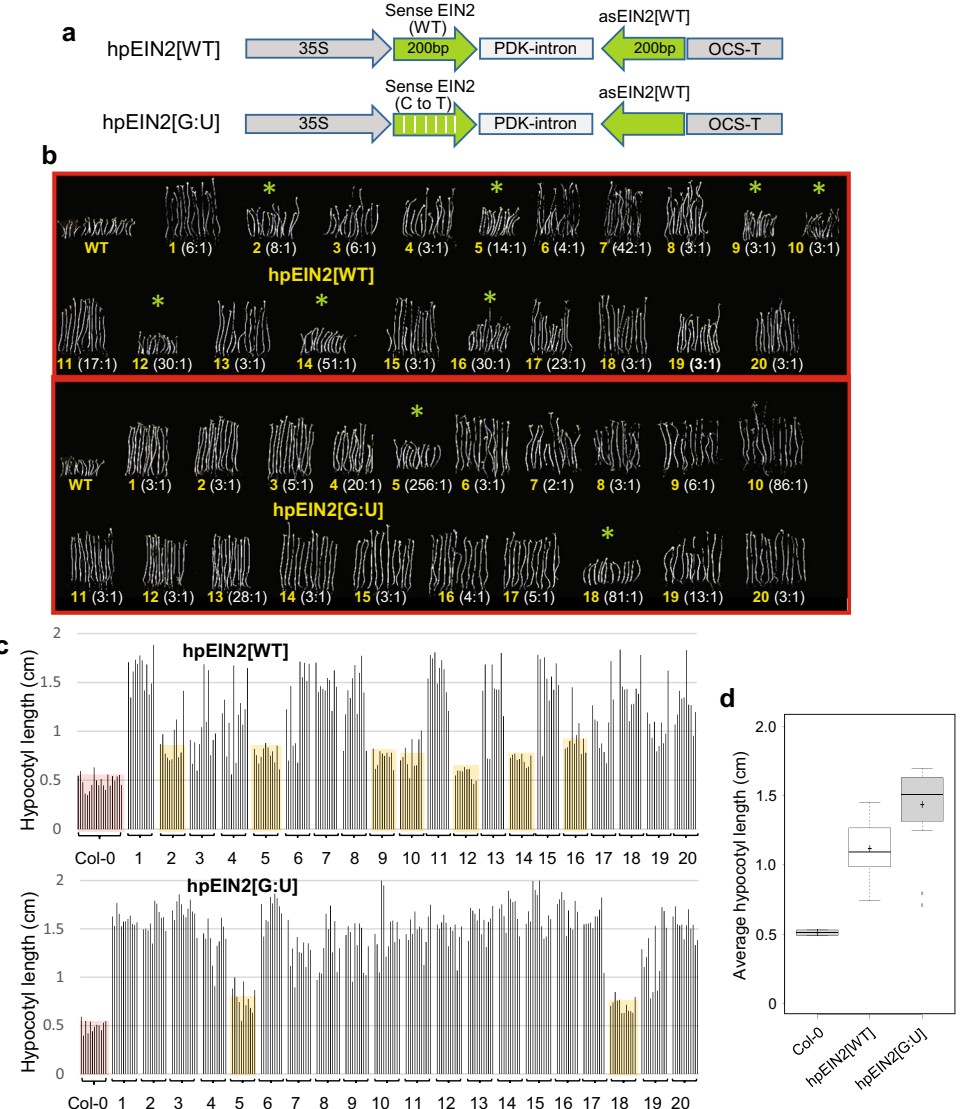

**Fig. 2 G:U modified hpRNA construct induces uniform RNAi of the endogenous *EIN2* gene in Arabidopsis. a** Schematic of the traditional (hpEIN2[WT]) and the G:U modified (hpEIN2[G:U] constructs. **b** Hypocotyl length phenotypes of 20 independent lines each of the two constructs, with longer hypocotyls indicating stronger *EIN2* RNAi. Approximately 12 T2 siblings with representative hypocotyl lengths were placed side by side and photographed. Untransformed (WT) Arabidopsis Col-0 plants were used as control. The kanamycin resistant:sensitive plant ratio for each line is shown in the bracket. **c** Measured lengths of the plants shown in (**b**). Plant lines with weak *EIN2* RNAi are highlighted with yellow shading. **d** Box plot (BoxPlotR, http://shiny.chemgrid.org/boxplotr/) of average hypocotyl lengths of the 20 independent hpEIN2[WT] and hpEIN2[G:U] lines in (**b**) and (**c**). Each data point corresponds to the average length of around 10-12 T2 siblings of an individual line ($n = 20$). For the WT Col-0 control, there are two data points ($n = 2$) containing 20 and 12 seedlings. The central horizontal line indicates the median value; the lower and upper borders of the box represent the first and third quartiles; whiskers extend 1.5 times the interquartile range from the first and third quartiles; outliers are represented by dots; crosses represent sample means. Source data are provided as a Source Data file.

only two very high-copy number lines (256:1 and 81:1) showing low levels of RNAi (Fig. 2b, c). Similar to a previous study[14], the hpEIN2[WT] lines with high-copy number insertions (with 8:1, 14:1, 30:1, 51:1 ratios) tended to show low levels of RNAi.

For the *PDS* target gene, we identified and analyzed 100 hpPDS[WT] and 172 hpPDS[G:U] primary transformants, all showing strong photo-bleaching in the cotyledons at the young seedlings stage (Fig. 3b; 7 days). Thus, both constructs induced effective *PDS* RNAi in cotyledons. However, the two populations showed a clear difference when true leaves emerged (14 days and beyond), with a much larger number of hpPDS[WT] plants giving green leaves that indicated a loss of strong RNAi (Fig. 3b). The hpPDS[G:U] population contained much higher proportions

of the strongly and moderately silenced lines (63 and 30% respectively) than the hpPDS[WT] population (34 and 23%) (Fig. 3c). In addition, most of the weakly silenced hpPDS[G:U] plants still showed mild mottling on true leaves, in contrast to the weakly silenced hpPDS[WT] plants that mostly had fully green leaves (Fig. 3b, c).

The *GUS*, *EIN2*, and *PDS* RNAi results collectively confirmed that the G:U hpRNA construct induces more uniform RNAi than the traditional hpRNA construct. Importantly, the *PDS* RNAi result indicated a developmental stage variability of RNAi by the traditional hpRNA transgene, being more effective in cotyledons than leaves, and suggested that the G:U hpRNA transgenes are developmentally more stable.

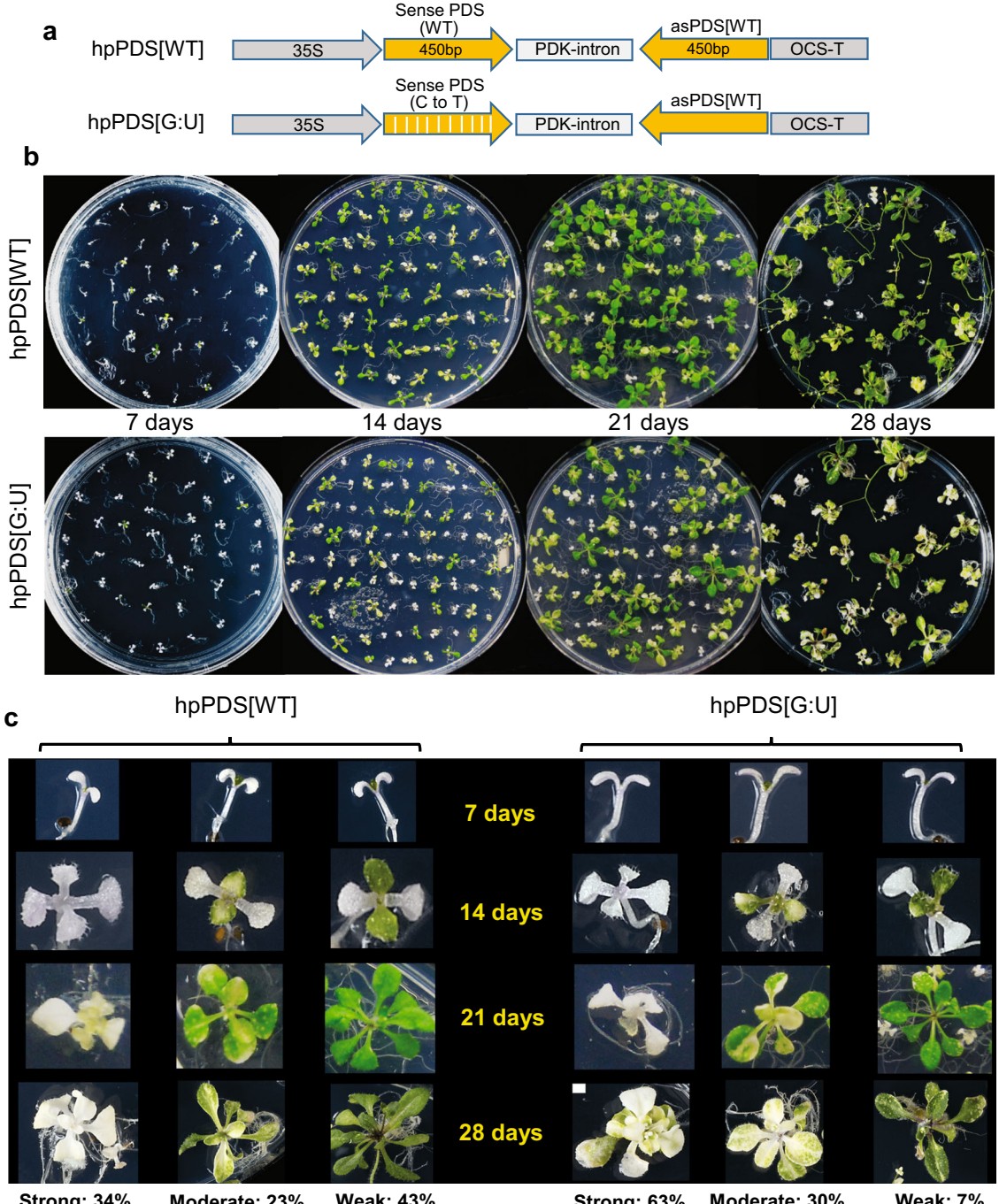

**Fig. 3 G:U modified hpRNA construct induces uniform and persistent RNAi of the endogenous *PDS* gene in Arabidopsis. a** Schematic of the traditional (hpPDS[WT]) and the G:U modified (hpPDS[G:U]) constructs. **b** Phenotypes of primary independent transformants with the hpPDS[WT] and hpPDS[G:U] constructs. Note that all lines have photo-bleached cotyledons, indicating strong *PDS* RNAi in cotyledons, but some lines developed green true leaves indicating loss of *PDS* silencing in leaves. **c** Summary of *PDS* RNAi in the primary transformants, showing a higher frequency of persistent *PDS* RNAi in the hpPDS[G:U] population. The plants were classified into three groups based on strong (strong photo-bleaching in the whole plant), moderate (pale green or mottled leaves), and weak (fully green or weakly mottled leaves) *PDS* RNAi.

**G:U hpRNA transgenes show diminished promoter methylation.** The enhanced uniformity of RNAi by the mismatched hpRNA transgenes suggested a reduced level of self-silencing, which could be due to reduced DNA methylation at the mismatched transgenes compared to the traditional hpRNA transgenes. To investigate this, we analyzed DNA methylation in the hpGUS, hpEIN2, and hpPDS transgenes, using methylation-dependent enzyme McrBC-digestion-PCR and bisulfite sequencing.

Seven of the 10 hpGUS[WT] lines analyzed (Fig. 4a) showed a clear reduction in PCR band intensity in McrBC-digested vs undigested samples (Fig. 4b), indicative of DNA methylation at the 35S-GUS junction region. These methylated lines included all the 5 lines (#2, 5, 8, 9, and 10) that showed no or low levels of GUS RNAi (Fig. 4a). In contrast to hpGUS[WT], all the 10 hpGUS[G:U] lines showed equal or near-equal PCR amplification between McrBC-treated and untreated samples (Fig. 4b), indicating no or low levels of DNA methylation. To more accurately

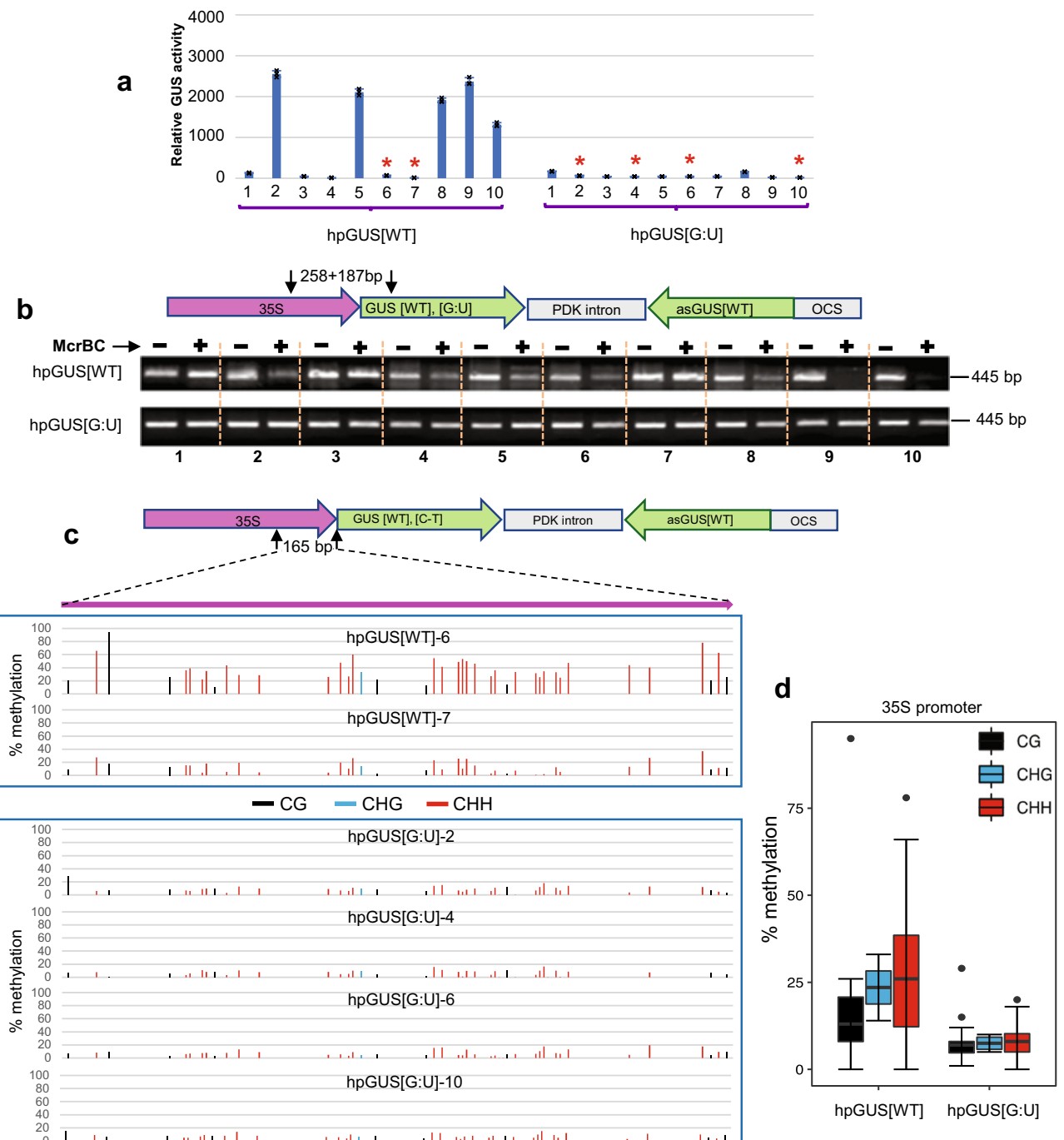

**Fig. 4 G:U modified hpRNA transgenes show reduced DNA methylation at the promoter region. a** GUS expression patterns of the independent transgenic lines were analyzed by McrBC-digestion PCR. Three technical replicates were measured for each line, and data are presented as mean values ± s.d. with the three data points shown as black cross. The red asterisks indicate the lines analyzed by bisulfite sequencing in (**c**). **b** McrBC-digestion PCR analysis of the 35S promoter-sense junction regions in the hpGUS transgenes as defined by the two arrows (the locations of forward and reverse PCR primers). McrBC is an endonuclease that only cleaves DNA containing methylated cytosine bases. Comparing PCR amplification of McrBC-treated with untreated DNA, therefore, provides an estimation of DNA methylation levels. DNA samples were either untreated (−) or treated (+) with McrBC before PCR amplification. **c** Bisulfite sequencing of the 35S promoter regions (the top strand only) as indicated by the two arrows (the locations of the nested forward and reverse bisulfite PCR primers). **d** Box plot (R version 3.6.0) showing the average CG, CHG, and CHH methylation levels for the 2 hpGUS[WT] and 4 hpGUS[G:U] lines in (**c**). The central horizontal line indicates the median value, the lower and upper borders of the box represent the first and third quartiles, and the outliers are represented by dots. Bisulfite treatment of genomic DNA converts unmethylated cytosine bases to uracil (U) (shown as thymine in PCR product) but methylated cytosines are not affected. PCR amplification of bisulfite-treated DNA followed by sequencing, therefore, detects methylated cytosines at single-nucleotide resolution. PCR primers were designed to specifically amplify only the 35S promoter sequences of the hpGUS transgenes but not the one driving HPT expression in the target GUS gene (Fig. 1a; 35S′). Source data are provided as a Source Data file.

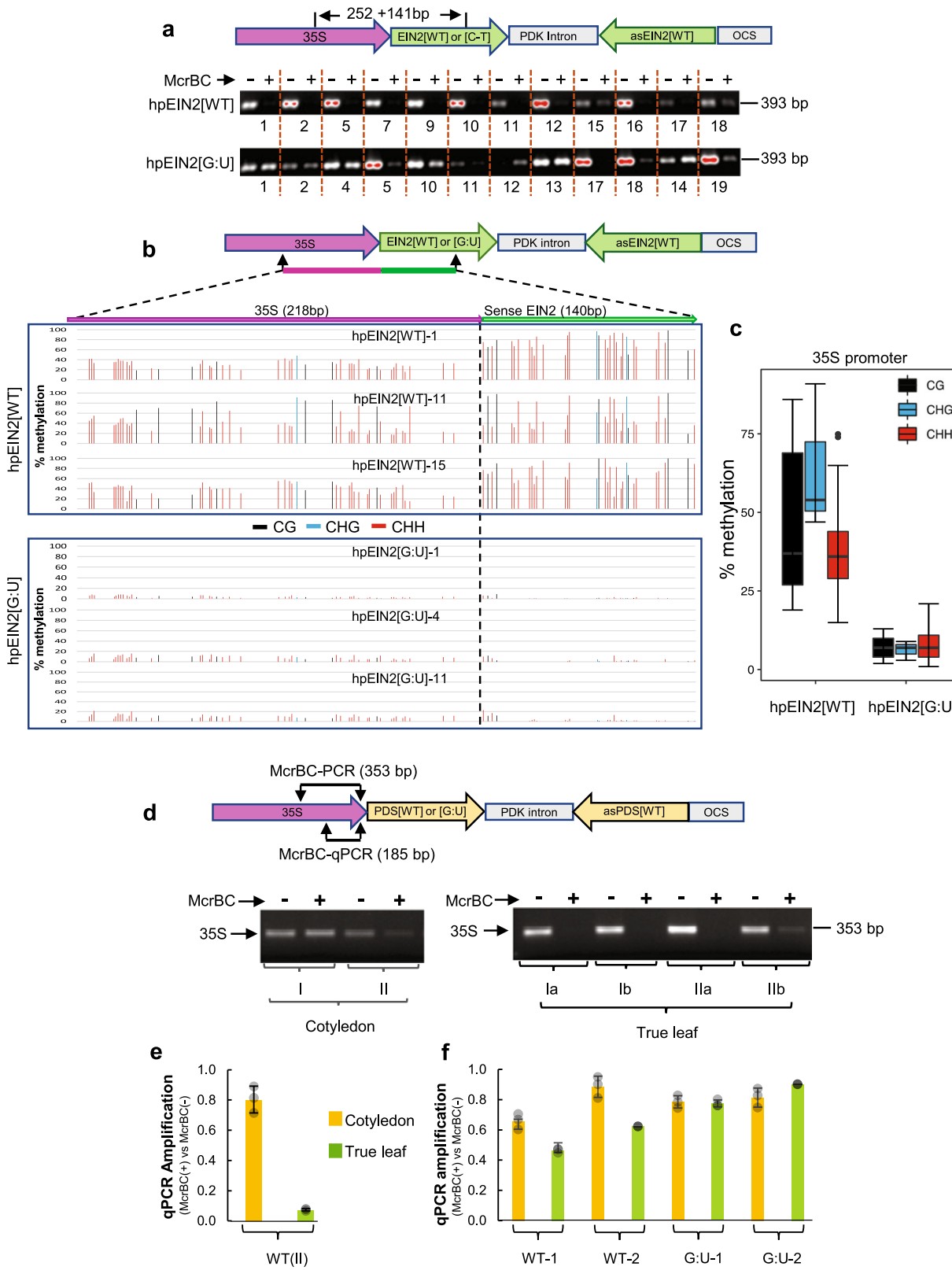

determine low DNA methylation levels, we performed bisulfite sequencing on two hpGUS[WT] and four hpGUS[G:U] lines that had strong RNAi (indicated by asterisks in Fig. 4a). Consistent with the McrBC-digestion PCR result, all four hpGUS[G:U] lines had very low levels of DNA methylation at the 35S promoter based on bisulfite sequencing (Fig. 4c, d). The two hpGUS[WT] lines, despite strong RNAi, both showed moderate levels of DNA

methylation in the 35S promoter (Fig. 4c, d). Thus, the hpGUS[G:U] transgene had reduced promoter methylation across the transgenic population compared to the hpGUS[WT] transgene.

Analysis of 12 independent hpEIN2[WT] and hpEIN2[G:U] lines each (Supplementary Fig. 4a) showed a clear difference in promoter methylation (Fig. 5). Every hpEIN2[WT] line had some

**Fig. 5 The G:U modified EIN2 hpRNA transgene shows greatly diminished DNA methylation at the promoter region. a** McrBC-digestion PCR of the 35S-EIN2 junctions regions (see Fig. 2b, S4a for *EIN2* silencing phenotypes). **b** Bisulfite sequencing of the 35S promoter-EIN2 junction region (the top strand only; the two arrows indicate the locations of the forward and reverse bisulfite PCR primers). Note that the 140 bp EIN2 sequence in hpEIN2[G:U] has no cytosines in the top strand so the low levels of signals reflect the background cytosine noises in the sequencing trace files. **c** Box plot (R version 3.6.0) showing the average cytosine methylation levels of the 35S promoter region in (**b**). The central horizontal line indicates the median value, the lower and upper borders of the box represent the first and third quartiles, and the outliers are represented by dots. **d**–**f** The hpPDS[WT] transgene shows stronger 35S promoter methylation in true leaves than in cotyledons. **d**, **e** McrBC-digestion PCR and qPCR of primary T1 transgenic lines. Primary T1 lines were randomly divided into two pools (I and II) and photo-bleached cotyledons from multiple T1 transgenic plants within each pool were collected and combined to generate two DNA samples. Young leaf tissues were also harvested from the same two groups but were divided into four pools (two pools for each of the two cotyledon pools) (Ia, Ib, and IIa, IIb). **f** McrBC-digestion qPCR of T2 transgenic plants. As T1 plants with strong *PDS* RNAi did not grow to seed, only two lines each with moderate *PDS* RNAi were analyzed. For each of the four lines, photobleached cotyledons and the first true leaves that had just emerged were harvested from ~25 T2 progeny, and used for DNA extraction and McrBC-digestion qPCR analysis. For qPCR in (**e**) and (**f**), three technical replicates were measured for each sample, and data are presented as mean values ± s.d. with the three data points shown as grey dots. Source data are provided as a Source Data file.

**Table 2 Summary of *PDS* and *EIN2* RNAi in RdDM mutant backgrounds.**

| Constructs | Total no. transgenic lines scored | Silenced lines | Unsilenced/weakly silenced lines |
| --- | --- | --- | --- |
| Col/hpPDS[WT] | 100 | 57 (57%) | 43 |
| Col/hpPDS[G:U] | 172 | 160 (93%) | 12 |
| *nrpd1a*/hpPDS[WT] | 52 | 46 (88.5%) | 6 |
| *nrpd1a*/hpPDS[G:U] | 40 | 40 (100%) | 0 |
| *ago4-2*/hpPDS[WT] | 59 | 56 (94.9%) | 3 |
| *ago4-2*/hpPDS[G:U] | 100 | 100 (100%) | 0 |
| Col/hpEIN2[WT] | 20 | 13 (65%) | 7 |
| Col/hpEIN2[G:U] | 20 | 18 (90%) | 2 |
| *nrpd1a*/hpEIN2[WT] | 20 | 20 (100%) | 0 |
| *nrpd1a*/hpEIN2[G:U] | 20 | 20 (100%) | 0 |
| *ago4-2*/hpEIN2[WT] | 19 | 16 (84.2%) | 3 |
| *ago4-2*/hpEIN2[G:U]* | 11 | 10 (90.9%) | 1 |

*A number of lines had poor germination on ACC medium in the darkness, possibly due to extreme *EIN2* RNAi, so the plant number is too small for proper comparison with the other transgenic populations.

levels of DNA methylation at the 35S-EIN2 junction, as indicated by the reduced PCR amplification of McrBC-digested samples (Fig. 5a). The widespread presence of promoter methylation in the hpEIN2[WT] lines was confirmed by bisulfite sequencing of three lines with the strongest *EIN2* RNAi hence least likely to be methylated (Supplementary Fig. 4a; #1, #11, #15). The 35S promoter region showed 20%~80% methylation for individual cytosines, with all three lines showing around 40% of cytosine methylation in the top strand of the sense *EIN2* sequence (Fig. 5b, c).

The hpEIN2[G:U] lines showed a clear reduction in promoter methylation, with 7 of the 12 lines showing little or no reduction in PCR amplification of McrBC-digested samples (Fig. 5a; hpEIN2[G:U] #1, 2, 4, 11, 12, 13, and 14). Bisulfite sequencing analysis confirmed the low methylation levels in the three hpEIN2[G:U] lines with strong *EIN2* RNAi (Supplementary Fig. 4a; #1, #4, #11), all showing less than 20% methylation for all cytosines in the top strand of 35S promoter (Fig. 5b, c). The two hpEIN2[G:U] lines that had weak RNAi (Fig. 2b, Supplementary Fig. 4a; #5 and #18) showed strong DNA methylation (Fig. 5a), indicating a direct link between the reduced *EIN2* RNAi and promoter methylation.

Many of the hpPDS[WT] lines showed strong *PDS* silencing in cotyledons but weak *PDS* silencing in leaves (Fig. 3), suggesting increasing promoter methylation from cotyledons to leaves. The majority of the hpPDS[G:U] lines exhibited strong *PDS* silencing phenotypes in both cotyledons and true leaves, indicating low promoter methylation levels in both tissues. McrBC-digestion-PCR detected a clear increase in DNA methylation at the 35S promoter of the hpPDS[WT] transgene in true leaves compared

to cotyledons (Fig. 5d–f), whereas promoter methylation remained similarly low in both tissues for the two hpPDS[G:U] lines based on McrBC-qPCR (Fig. 5f).

Taken together, the methylation analyses indicated that the relatively uniform RNAi of the mismatched hpRNA lines was due to diminished promoter methylation and that the traditional hpRNA transgenes are inherently prone to promoter methylation with all lines having some levels of promoter methylation. The results also suggested that promoter methylation of traditional hpRNA transgenes is developmental stage-dependent.

**Methylation of hpRNA transgenes is retained in RdDM mutants.** It was thought that the methylation in the IR region of a traditional hpRNA transgene is induced by hpRNA-derived siRNAs via the RdDM pathway. Consequently, it was expected that the traditional hpRNA transgenes would lose the methylation in an RdDM mutant resulting in uniform RNAi across transgenic populations. It was also expected that the traditional hpRNA transgenes would induce more effective RNAi than the G:U hpRNA transgenes in RdDM mutants due to stronger dsRNA stability. We investigated these using two Arabidopsis RdDM mutants, *nrpd1a-3* (a T-DNA insertion mutant of the upstream siRNA biogenesis factor Pol IV) and *ago4-2* (a mutant of the downstream effector AGO4).

The traditional hpRNA constructs, targeting *PDS* or *EIN2*, indeed induced uniform RNAi in the two RdDM mutants, with 84~100% of transgenic lines showing RNAi (Table 2). The white cotyledon-to-green leaf-type of *PDS* RNAi phenotype of the Col-0

background (Fig. 3) also largely disappeared in the RdDM mutants, with most of the hpPDS[WT] plants showing relatively uniform photo-bleaching from cotyledons to leaves (Supplementary Figs. 5a, 6a). However, to our surprise, the traditional hpRNA transgenes induced weaker RNAi than the G:U transgenes in both mutant backgrounds (Supplementary Fig. 5; Supplementary Fig. 6a). In particular, the hpPDS[G:U] construct induced extreme photo-bleaching in 100% of the transgenic lines in *ago4-2*, compared with moderate to high levels of photo-bleaching in 95% of the hpPDS[WT] lines (Supplementary Fig. 5a; Supplementary Fig. 6a). The RNAi of *ago4-2*/hpEIN2[G:U] lines could not be properly assayed using hypocotyl length because many lines showed poor seed germination in the dark on ACC medium (likely due to strong *EIN2* RNAi). Nevertheless, under light, the T2 plants of hpEIN2[G:U] lines showed more vigorous root and foliage growth, particularly in the *ago4-2* mutant (Supplementary Fig. 5b), indicating stronger RNAi than the hpEIN2[WT] lines. It is worth noting that in the *nrpd1a-3* mutant, no hpPDS[WT] lines developed strong photo-bleaching, indicating that the extent of *PDS* RNAi was reduced in this Pol IV mutant despite the increased uniformity of RNAi across transgenic populations.

The hpEIN2[WT] transgenes developed similarly strong DNA methylation in the IR and promoter regions (top strand) in the RdDM mutants as in the wild-type Col-0 background, even in the lines with strong *EIN2* RNAi (Fig. 6). In contrast, DNA methylation was largely absent in the hpEIN2[G:U] transgenes in all backgrounds (except for *nrpd1a*/hp[G:U]−13). Similarly, strong methylation in the IR region (the sense *PDS* sequence) was also retained in the hpPDS[WT] lines in the *ago4-2* and *nrpd1a-3* mutants (Supplementary Fig. 6b). Thus, strong DNA methylation inside the perfect inverted-repeat DNA, as well as its spread to the upstream promoter, was retained in the two mutants of the RdDM components.

The *EIN2* and *PDS* genomic targets (top strand) also showed DNA methylation but at a much lower level than the IR region of the hpEIN2[WT] and hpPDS[WT] transgenes, particularly at the CHG and CHH sites (Fig. 6; Supplementary Fig. 6b; Supplementary Fig. 7) (Note that the four CG sites near the 5′ half of the *EIN2* target were already heavily methylated in untransformed Col-0, *ago4-2*, and *nrpd1a-3* plants; Supplementary Fig. 8). Unlike the hpRNA transgenes, target gene methylation was clearly reduced in the *nrpd1a-3* mutant for the hpRNA[WT] lines, suggesting that Pol IV, the siRNA biogenesis component of the canonical RdDM pathway, is involved in hpRNA-induced target gene methylation. Remarkably, the G:U hpRNA transgenes induced similar levels of non-CG methylation at the target gene loci (top strand) in both the WT Col-0 and *nrpd1a-3* backgrounds (Fig. 6; Supplementary Fig. 6b; Supplementary Fig. 7), suggesting that the G:U hpRNA transgene-induced RdDM was less dependent on Pol IV than the traditional hpRNA-induced RdDM.

The bottom strand of *EIN2* genomic target showed strongly reduced DNA methylation than the top strand in the hpEIN2[G:U] plants of Col-0 and *nrpd1a-3* backgrounds, particularly of the *nrpd1a-3* background (Supplementary Fig. 8). This strand bias suggested that RdDM requires strong sequence complementarity between siRNAs and target DNA and that the sense siRNAs from primary G:U hpRNA transcript were unable to efficiently induce methylation at the target DNA due to nucleotide mismatches with the bottom DNA strand.

Taken together, experiments with the RdDM mutants further indicated that DNA methylation of traditional hpRNA transgenes is intrinsic to the IR DNA structure which persisted in mutants of both the upstream (Pol IV) and downstream (AGO4) RdDM components. This intrinsic methylation prevented the traditional

hpRNA transgenes from reaching their full RNAi efficacy, even in these RdDM mutants. However, the increased cross-line uniformity of *PDS* and *EIN2* RNAi in the *ago4-2* and *nrpd1a* mutants suggested that these RdDM components contribute to genomic position or copy number-dependent silencing of hpRNA transgenes.

**Traditional and G:U hpRNAs are differently processed.** One obvious question was whether G:U base-paired hpRNAs were efficiently processed by Dicer into siRNAs. Northern blot analysis detected abundant siRNAs from the hpEIN2[G:U] (Fig. 7a) and hpGUS[G:U] (Supplementary Fig. 9a) plants. The amount of siRNAs looked more even across the independent G:U hpRNA lines than the traditional hpRNA lines, and showed a good correlation with the extent of RNAi (Supplementary Fig. 4a, b). Thus, the uniform RNAi across independent G:U hpRNA lines could be attributed to relatively even amounts of siRNAs. The strong RNAi lines of hpEIN2[G:U] accumulated similar amounts of siRNAs to the strong RNAi lines of hpEIN2[WT] (except for hpEIN2[WT]−17), indicating that G:U hpRNA was efficiently processed by Dicer. The strong hpGUS[WT] lines, however, accumulated much higher amounts of siRNAs than the hpGUS[G:U] lines (Supplementary Fig. 9a). *GUS* target mRNA, from a highly expressed transgene, may serve as the template for the production of secondary siRNAs by RDR[16], but the low numbers of GUS siRNAs from downstream of the targeted region in the hpGUS[WT] plants (Supplementary Fig. 10) implied that target RNA-derived secondary siRNAs were not a major contributor to the amounts of hpGUS[WT] siRNAs.

siRNAs from both traditional and G:U hpRNA forms two dominant bands on the gel, consistent with hpRNA transgenes generating primarily 21 and 24 nt siRNAs[17]. However, the two siRNA bands of hpEIN2[G:U] (and hpGUS[G:U]) plants showed faster gel mobility (Fig. 7a, b; Supplementary Fig. 9a). Small RNA deep sequencing (sRNA-seq) detected no clear shift in siRNA size profiles between the traditional and G:U hpRNA lines, with the 21 nt siRNAs being always the dominant followed generally by the 24 nt or 22–23 nt siRNAs (Fig. 8; Supplementary Fig. 9b). The antisense siRNAs in the hpEIN2[G:U] lines are less abundant than the sense, G:U modified siRNAs (Fig. 8). It is possible the sense siRNAs were relatively enriched for 5′ U and therefore preferentially loaded to AGO1 resulting in higher abundance than the antisense siRNAs.

The gel mobility difference of siRNAs between traditional and G:U hpRNA plants prompted us to investigate if they possess different chemical modifications at the 5′ and 3′ termini. Plant siRNAs are generally methylated at the 3′ terminus[7], and in accordance with this, a β-elimination assay indicated that siRNAs from both hpEIN2[WT] and hpEIN2[G:U] were 3′-methylated (Fig. 7c). Dicer-processed sRNAs were assumed to have 5′ monophosphate but in *C. elegans* many siRNAs are found to possess di- or tri-phosphate which increases gel mobility in denaturing polyacrylamide gels with high acrylamide:bis-acrylamide ratios[18]. Alkaline phosphatase treatment reduced the gel mobility for both hpRNA[WT] and hpRNA[G:U]-derived siRNAs (Fig. 7d), which migrated at more similar positions than without the phosphatase treatment. This raises the possibility that the two siRNA populations may have different 5′ phosphorylation. The siRNA bands of hpEIN2[WT] plants aligned well with the 21 and 24 nt sRNA size markers that were monophosphorylated with radioactive $^{32}$P (Fig. 7a), suggesting that these siRNAs are largely monophosphorylated. The G:U hpRNA-derived siRNAs, with faster mobility, could therefore possess 5′ di- or multi-phosphate. This possibility was supported by the northern blot showing that the di- and tri-phophorylated sRNA

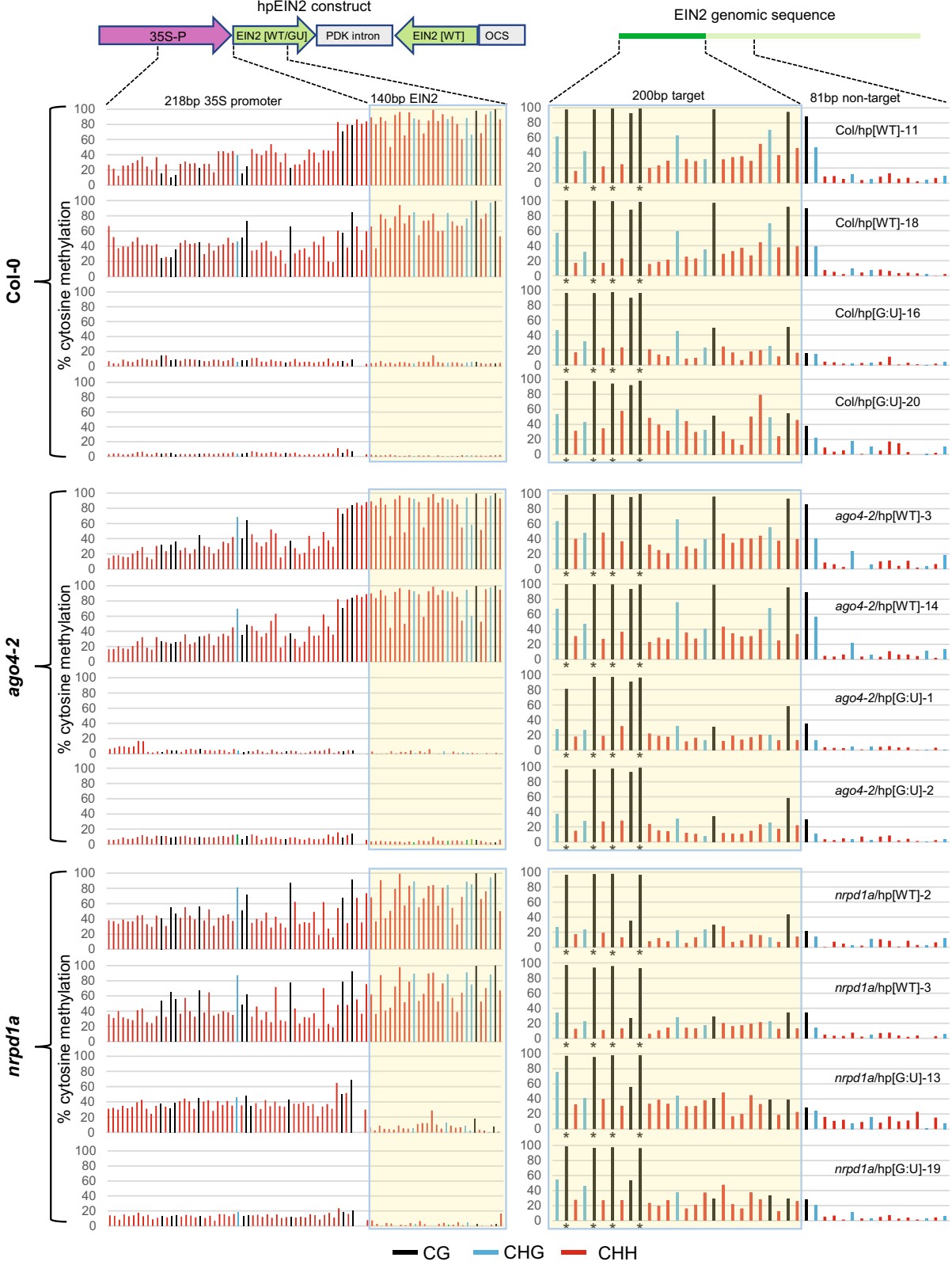

makers migrated faster than the unphosphorylated or mono-phosphorylated markers, and that the hpEIN2[G:U]-derived 24-nt siRNAs migrated at a closer position to the tri and di-phosphorylated 24 nt sRNA markers (Fig. 7e; Supplementary Fig. 11). Northern blot hybridization detected high amounts of long dsRNA species in the hpEIN2[G:U] lines but not in the

hpEIN2[WT] plants (Fig. 7b), suggesting that the two types of hpRNA are processed differently. The Arabidopsis microRNA miR168 showed a smaller gel mobility shift than the trans-acting siRNA tasiR255 after alkaline phosphatase treatment (Fig. 7d), suggesting that plant endogenous sRNAs may also possess variable 5′ phosphorylation.

**Fig. 6 Heavy DNA methylation in the traditional hpEIN2[WT] transgene is retained in AGO4 mutant *ago4-2* or the PolIV mutant *nrpd1a-3*.** Yellow-highlighted areas represent the IR (left) and target *EIN2* genomic (right) regions. Left, both promoter and IR DNA methylation of hpEIN2[WT] is retained in the *ago4-2* and *nrpd1a-3* mutants, with hpEIN2[G:U] showing almost no methylation in the 35S promoter except for the *nrpd1a-3*/hp[G:U]-3 line. Note that only the top strand was analyzed; the 140 bp *EIN2* sequence in hpEIN2[G:U] has no cytosines in the top strand so the low levels of signals reflect the background cytosine noises in the sequencing trace files. Right, methylation levels of the hpRNA-targeted *EIN2* genomic sequence (top strand only), show much lower levels of CHG and CHH methylation than the hpEIN2[WT] sequence on the left, particularly in the *nrpd1a*/hpEIN2[WT] lines. The asterisks underneath the graph indicate the CG sites that were already densely methylated in the untransformed Col-0, *ago4-2*, and *nrpd1a* backgrounds (Supplementary Fig. 8). Source data are provided as a Source Data file.

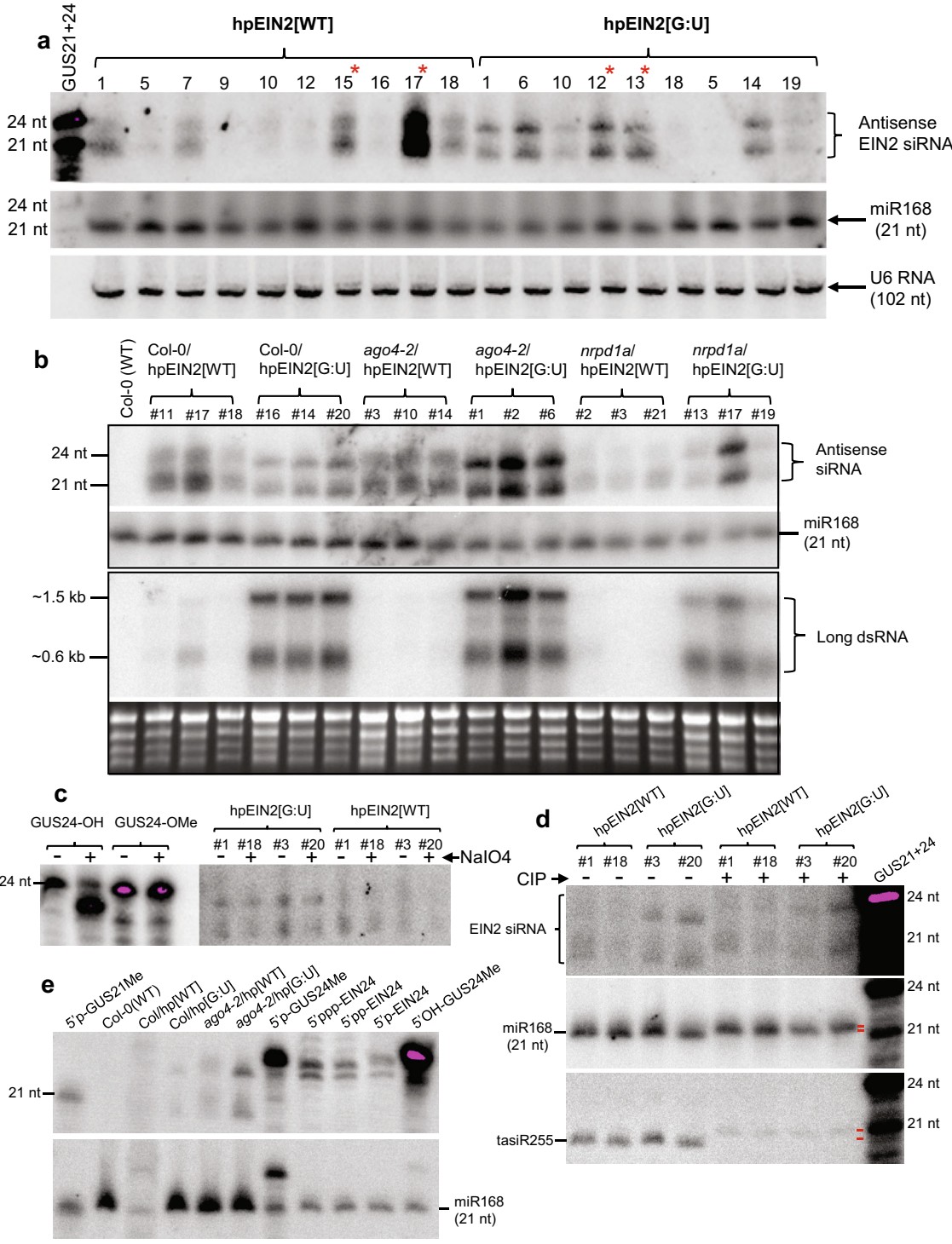

**Fig. 7 G:U hpRNAs are differently processed from traditional hpRNAs. a** Northern blot hybridization to detect antisense sRNAs from T2 hpEIN2[WT] and hpEIN2[G:U] Arabidopsis plants using sense RNA transcripts of the 200 bp *EIN2* target sequence as probe (see Fig. 2 and Supplementary Fig. 4a for RNAi phenotypes). Asterisks indicate samples for sRNA deep sequencing. **b** Detection of antisense sRNAs (upper panel) and long dsRNA (lower panel) using the same *EIN2* probe as in (**a**). Ethidium bromide-stained rRNA is used as the loading control. **c** β-elimination (NaIO4 treatment) assay confirming similar 3′-O-methylation between hpEIN2[WT] and hpEIN2[G:U]-derived siRNAs (If hydroxyls on the 3′-terminal ribose are unmethylated, NaIO4 oxidizes them to form an unstable dialdehyde that leads to β-elimination of the terminal nucleoside and an approximately 2-nt downward mobility shift). **d** Alkaline phosphatase (CIP) treatment of sRNAs. The same gel blot was sequentially hybridized with the sense *EIN2*, trans-acting siRNA255 (tasiR255), and miR168 probes. Note that CIP treatment resulted in slowed but more similar siRNA gel migration between the two hpRNA designs. Also, note that tasiR255 showed a greater gel mobility shift than miR168 (see the different gaps between the two short red lines that indicate the average position of the CIP-treated and untreated tasiR255 and miR168 bands. It was unclear why the intensity of tasiR255 band was markedly reduced upon CIP treatment). **e** Detection of hpEIN2[WT] and hpEIN2[G:U]-derived siRNAs together with 5′ labeled 21 and 24 nt (5′p-GUS21Me, 5′p-GUS24Me) or un-labeled 24 nt (5′ OH-GUS24Me) synthetic GUS sRNA markers (Supplementary Data 1), and in vitro transcribed mono- (5′P), di- (5′PP) and tri- (5′PPP) phosphorylated *EIN2* sRNA markers (see Methods). All marker samples were mixed with 2 μg total RNA of WT Col-0 before loading. The un-labeled GUS24 and *EIN2* sRNA markers were visualized by hybridization with respective antisense oligonucleotide probes. Note that the gel migration of 24 nt sRNA markers was slowest for the non-phosphorylated 5′OH-GUS24Me and the fastest for the di- and tri-phosphorylated *EIN2* markers. The Col/hp[WT] sample looked degraded on this gel but another gel was run to verify the sRNA pattern (Supplementary Fig. 11b). Source data are provided as a Source Data file.

## Discussion

In this study we showed that the traditional hpRNA transgenes are invariably methylated at the IR DNA structure and the adjacent promoter sequences compromising RNAi efficiency. This widespread intrinsic DNA methylation and self-silencing of hpRNA transgenes were not reported before but is nevertheless unsurprising. IR DNA structures have long been reported to attract DNA methylation that can extend short distance to upstream promoters in plants, and the methylated IR locus can induce homology-dependent trans-methylation of single-copy loci in the genome[19–21]. The best-studied IR DNA is the naturally occurring *PAI1-PAI4* locus in Arabidopsis ecotype Wassilewskija, which always carries dense DNA methylation independently of its transcriptional activity or RdDM factors[22]. Evidence exists that supports DNA:DNA pairing in IR-induced methylation, but dsRNA and sRNA signals are also suggested to contribute to the methylation, particularly at the homologous trans-methylated non-IR loci[20,23]. Our results showed that strong DNA methylation in the hpRNA transgenes was largely retained in the mutants of both the upstream siRNA biogenesis factor Pol IV (*nrpd1a-3*) and the downstream effector AGO4 (*ago4-2*) of RdDM, which seems to support a RdDM-independent DNA:DNA pairing model in IR methylation. However, the increased uniformity of RNAi across transgenic populations in the *ago4-2* and *nrpd1a-3* mutants by the traditional as well as G:U hpRNA transgenes suggest that hpRNA transgenes, like any type of transgenes, are also subject to insertion pattern or position-dependent transcriptional silencing, and that RdDM plays a key role in this type of transgene silencing.

It is interesting to note that RNAi potency was generally reduced in the *nrpd1a-3* mutant compared to wild-type Col-0 and the *ago4-2* mutant, as indicated by the uniform but weak photobleaching phenotypes of hpPDS lines and the low amount of hpEIN2-derived siRNAs in the *nrpd1a* background. The *nrpd1a-3* mutant contained a T-DNA insertion with a 35S promoter enhancer sequence that could cause transcriptional silencing to the 35S promoter driving hpRNA expression[24]. However, the detection of relatively high-abundance long dsRNA intermediates as well as siRNAs in all the three *nrpd1a*/hpEIN2[G:U] lines (Fig. 7b) suggested that substantial trans-inactivation of the 35 S promoter did not occur in the *nrpd1a* background. It has been proposed previously that Pol IV may use either methylated DNA and/or dsRNA as templates to generate dsRNA and siRNAs[25]. More direct evidence for the dsRNA-templated model came from a study showing that RNA virus-derived siRNAs, with no DNA source, are strongly reduced in a Pol IV mutant[26]. The hpEIN2[G:U] plants accumulated high amounts of long dsRNA

species, and the bulk of sense siRNAs had the C to U-modified sequence, indicating that siRNAs were mostly derived from direct Dicer processing of the primary G:U hpRNA transcript independent of Pol IV. For the hpEIN2[WT] transgenes, however, long dsRNA was almost undetectable and there was a strong reduction in siRNA accumulation in the *nrpd1a-3* background (Fig. 7b). This raises the possibility that Pol IV may contribute specifically to siRNA production from the traditional hpRNA transgenes using the low amounts of the primary perfect dsRNA as a template. Interestingly, siRNA bands of hpRNA[WT] looked more scattered on the gel blot than those of hpRNA[G:U] (Fig. 7), which implies that hpRNA[WT]-derived siRNAs are a mixture of different biogenesis processes with different size or 5′ phosphorylation hence different gel mobility (e.g. direct Dicer processing of primary hpRNA plus Pol IV-mediated amplification), unlike the G:U hpRNA-derived siRNAs that are largely derived from the primary hpRNA transcript.

The key finding of this study is that C to T substitutions or around 25% nucleotide modifications in the sense DNA sequence prevented the intrinsic methylation of the hpRNA transgenes, resulting in uniform RNAi across independent transgenic lines. The C to T substitutions also prevented the cotyledon to true leaf progression of methylation and self-silencing observed for the hpPDS[WT] transgene, a phenomenon that has not been reported before but has important implications in studying developmental stage-dependent RNAi and transcriptional gene silencing. Thus, disruption of perfect IR DNA structures is sufficient to block IR methylation and self-silencing of hpRNA transgenes. It is interesting to note that microRNA precursors in plants usually contain mismatches or G:U base-pairs in the duplex regions. Considering the results from our study, this structural feature may have evolved to disrupt IR DNA structure preventing transcriptional self-silencing of miRNA genes.

As illustrated by the different *GUS* RNAi efficacy by the four hpGUS constructs, reduced dsRNA stability due to nucleotide modifications in the sense strand reduces RNAi efficiency presumably because of inefficient Dicer processing. Weak to moderate RNAi can have specific applications, particularly when the target genes are required for plant viability. The potential drawback of reduced RNAi, however, is largely overcome with the G:U hpRNA constructs, where the C-to-T changes in the sense sequence disrupt the IR DNA structure but still allow the formation of perfect hpRNA structure due to G:U wobble base-pairing. Consequently, all three G:U hpRNA constructs tested induced strong and uniform RNAi. hpRNAs containing multiple G:U base-pairs (up to 17.5%) has been previously shown to induce RNAi in animals and confer virus resistance in plants[27,28].

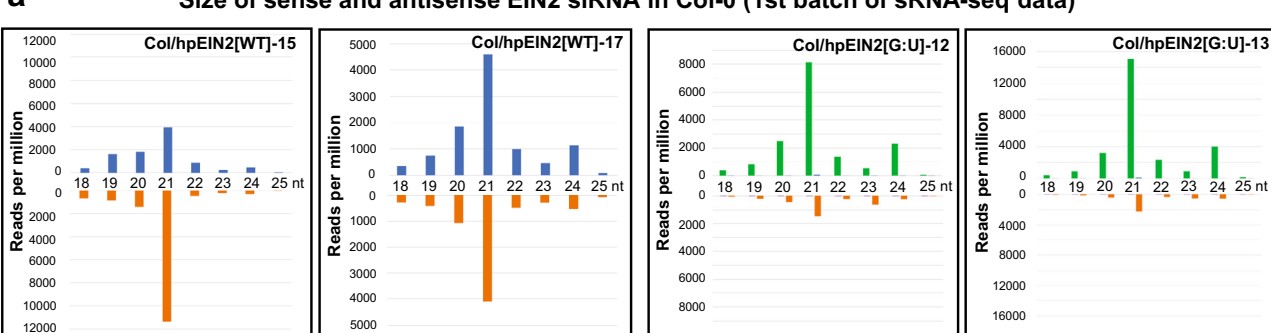

**Fig. 8 Summary of sRNA deep sequencing data.** Size distribution of sense and antisense siRNAs derived from the dsRNA stem of the hpEIN2[WT] and hpEIN2[G:U] in Col-0 (**a**, **b**), *nrpd1a-3* and *ago4-2* (**b**) backgrounds is shown. Note that for the G:U hpRNA lines, the bulk of the sense siRNAs have the C to U converted sequence of the sense strand, indicating little or no RDR-synthesized secondary siRNAs.

In our study all cytosines in the sense sequence, constituting 18~26% of the target sequences, were substituted in the G:U hpRNA constructs. Future studies should examine the number of C-to-T substitutions that are required for reducing self-silencing while maximizing RNAi efficiency.

Our study indicated that G:U hpRNA is differently processed compared to the traditional hpRNA. The hpEIN2[G:U] lines accumulated high amounts of distinct large-sized dsRNA intermediates, which were largely absent in the hpEIN2[WT] lines. Furthermore, while the hpEIN2[G:U] plants accumulated similar amounts of siRNAs to the hpEIN2[WT] lines (Fig. 7b), siRNAs from the two hpRNA designs showed different gel mobilities.

Alkaline phosphatase treatment homogenized the gel mobility of the two siRNA populations, raising the possibility that siRNAs from the G:U and traditional possess different 5′ phosphorylation. Similarly, the endogenous miR168 and tasiR255 sRNAs also showed different gel mobility and alkaline phosphatase-caused gel mobility shifts, suggesting that endogenous plant sRNAs could also have different 5′ phosphorylation. Methylation analysis of hpRNA transgenes in the *nrpd1a-3* and *ago4-2* mutants suggested that the G:U hpRNA-derived siRNAs, unlike those of the traditional hpRNA, induce RdDM through a Pol IV-independent pathway. Thus, G:U hpRNA-derived siRNAs may have distinct functional properties from the traditional hpRNA-derived

siRNAs, possibly due to different biogenesis or 5′ modification. Further studies are needed to confirm and understand any difference in chemical modification of traditional and G:U hpRNA-derived siRNAs.

In conclusion, our study uncovered a RNAi construct design that overcomes transcriptional self-silencing to induce more uniform and persistent RNAi than the traditional hpRNA design and shed light on IR DNA-induced gene silencing in plants. Apart from theoretical interest, future studies should investigate if G:U-modified and other mismatched hpRNA transgenes also have increased long-term stability inducing effective RNAi in multiple generations, which would be important for field applications of RNAi in crop improvements.

## Methods

**Plant materials and growth conditions**. Plants used in the experiments included *Arabidopsis thaliana* (ecotype Col-0), and transgenic *Nicotiana tabacum* Wisconsin 38 lines PPGH11 and PPGH24. These are two independent lines homozygous for the single-copy transgene expressing *GUS* driven by a promoter from the *Cucurbita pepo* PP2 gene[29]. The PP2-GUS plants were chosen as the testing plants because the PP2 promoter came from an endogenous gene with a different sequence to the 35S promoter used to drive the expression of the hpRNA transgenes, which therefore would prevent transcriptional silencing of the target GUS gene by promoter trans-inactivation. Plant seeds were sown either directly into soil, or placed first on MS plate for germination followed by transferring seedlings to soil. Plants were grown in a growth room (16 h light/8 h dark) at 22–24 °C.

**Construct preparation**. For preparing GUS hpRNA constructs, the 200 bp *GUS* ORF sequence (nt. 801–1000 from the translational start codon ATG) was PCR-amplified using the oligonucleotide primer pair GUS-WT-F and GUS-WT-R (Supplementary Table 1), containing *Xho*I and *Bam*HI sites or *Hin*dIII and *Kpn*I sites, respectively. PCR fragment was inserted into pGEM-T Easy (Promega Cat No. A1360), the correct nucleotide sequence confirmed by sequencing, and inserted as a *Bam*HI/*Hin*dIII fragment into pKannibal[30] forming the 35S-P::PDK intron::antisense GUS::Ocs-T cassette (pMBW606). This plasmid was used as the base vector for assembling the four GUS hpRNA constructs as follows.

For making hpGUS[WT], the 200 bp *GUS* PCR fragment was excised from the pGEM-T Easy plasmid with *Xho*I and *Kpn*I, and inserted into the same sites in pMBW606 between the 35S promoter and the PDK intron. For making hpGUS[1:4], hpGUS[2:10] and hpGUS[G:U], the 200 bp 1 in 4 mismatched, 2 in 10 mismatched and C to T modified sequences were assembled by annealing the respective pair of overlapping oligonucleotides (GUS-4M-F + GUS-4M-R for GUS[1:4], GUS-10M-F + GUS-10M-R for GUS[2:10], and GUS-GU-F + GUS-GU-R for GUS[G:U]; Supplementary Table 1) followed by PCR extension of 3′ ends using the high fidelity LongAmp Taq polymerase (NEB Cat No. M0323S). Nucleotide substitutions in GUS[1:4] and GUS[2:10] followed the following rule: C is changed to G, G to C, A to T and T to A. The PCR fragments were ligated into the pGEM-T Easy vector, the correct nucleotide sequences confirmed by sequencing, and then inserted as a *Xho*I/*Kpn*I fragment into pMBW606. The resulting 35S promoter::hpRNA::OCS terminator cassette was excised with *Not*I and inserted into the *Not*I site of pART27[31], forming the four final hpGUS constructs.

For preparing the traditional and G:U base-paired *EIN2* and *PDS* hpRNA constructs, DNA fragments spanning the 200 bp regions of the wild-type *EIN2* cDNAs were PCR-amplified from *Arabidopsis thaliana* Col-0 cDNA using the oligonucleotide primer pairs EIN2wt-F and EIN2wt-R (Supplementary Table 1) and cloned into pGEM-T Easy. The 200 bp C to T modified sense sequence (EIN2[G:U]) was assembled by annealing the overlapping oligonucleotides EIN2-GU-F and EIN2-GU-R (Supplementary Table 1), followed by PCR extension of 3′ ends using LongAmp Taq polymerase, and also cloned into pGEM-T Easy and sequenced. DNA fragments of 450 bp wild-type and C-to-T modified sequences of *PDS* cDNA (Supplementary Table 2) were synthesized by GeneArt™.

The 35S-P::sense fragment::PDK intron::antisense fragment::OCS-T cassettes were prepared in the same way as for the hpGUS constructs. Essentially, the wild-type sequences were excised from the respective pGEM-T Easy plasmids by digestion with *Hin*dIII and *Bam*HI, and inserted into pKannibal between the *Bam*HI and *Hin*dIII sites so they would be in the antisense orientation relative to the 35S promoter. The wild-type or C to T modified fragments were then excised from the respective plasmids using *Xho*I and *Kpn*I and inserted into the same sites of the respective antisense-containing clone. All of the cassettes in the pKannibal vector were then excised with *Not*I and inserted into pART27 to form the final binary vectors for plant transformation.

**Stable transformation and identification of transgenic lines**. All four *GUS* hpRNA constructs were transformed into the GUS-expressing tobacco lines PPGH11 and PPGH22 using the *Agrobacterium*-mediated leaf-disk method[32].

*EIN2* and *PDS* hpRNA constructs were transformed into *A. thaliana* by the floral dipping method[33]. To select for transgenic Arabidopsis lines, mature seeds were sterilized[34] and spread on MS plates containing 50 μg/mL kanamycin (Sigma Aldrich Cat No. K1377) plus 150 μg/mL timentin (Fisher Scientific Cat No. NC9588884) to inhibit Agrobacterium growth. The phenotype of *PDS* silencing was recorded for the primary (T1) transformants. The surviving T1 lines of *PDS* hpRNA constructs, and those of *EIN2* hpRNA construct, were transferred to soil, self-fertilized and grown to maturity. Seed collected from these plants (T2 seed) was used to establish T2 plants that were used for further gene silencing, DNA methylation, and transgene segregation analyses.

**Analysis of *GUS* and *EIN2* silencing**. GUS activity was quantitatively determined using fluorimetric 4-methylumbelliferyl-β-D-glucuronide (Merck Cat No. 89105) (MUG) assay[34]. The relative GUS activity represents the slope value per 5 mg of protein. For T0 plants (the primary transformants), protein used for the MUG assay was extracted from 3 leaves of an individual plant, while for the second generation, the protein was extracted from a pool of multiple (20–50) transgenic plants.

For *EIN2* silencing assay, Arabidopsis seeds were sterilized[34] and plated on half-strength MS salt medium (Sigma Aldrich Cat No. M5519) (without organics) containing 5 mg/L 1-aminocyclopropane-1-carboxylic acid (Sigma Aldrich Cat No. 149101-M) (ACC). The plates were imbibed for 3 days at 4 °C in the dark, transferred to 22 °C under lights for 10 h to improve germination, and then incubated for 4 days in the dark. Around 10–12 seedlings from each transgenic line, representing the overall hypocotyl length distribution, were selected from the half-strength MS salt medium and positioned horizontally onto agar plates containing blue stain to visualize hypocotyl length. The hypocotyl length of the seedlings was photographed using a digital camera and measured using ImageJ (http://rsb.info. nih.gov/ij).

**DNA and RNA analysis**. DNA, small RNA and large RNA from all transgenic tobacco lines were prepared following the phenol extraction method[10]: plant tissues were ground to powder in liquid nitrogen, and suspended quickly in pre-heated phenol:RNA extraction buffer (100 mM LiCl, 1% SDS, 100 mM Tris pH9, 10 mM EDTA) (1:1 ratio). An equal volume of chloroform was added and mixed, the mixture was centrifuged and the supernatant was transferred to a new tube. Lithium chloride was added to the supernatant at a 2 M final concentration, and large RNA precipitated at 4 °C overnight. Supernatant from large RNA precipitation was then mixed with 1 volume of isopropanol to precipitate DNA and small RNA. Total RNA from the T2 transgenic Arabidopsis lines was extracted using TRIzol® Reagent (Ambion® USA Thermofisher Cat No. 15596018) according to the manufacturer's instructions. The genomic DNA from the T2 transgenic plants was isolated from plant leaves using a Cetyltrimethyl Ammonium Bromide (CTAB) (Sigma Aldrich cat No. H6269) method[35].

For Southern blot hybridization, 10 μg of genomic DNA was digested with *Hin*dIII enzyme at 37°C, separated in 0.8% agarose gel, and blotted to HyBond-N$^+$ membrane (GE Healthcare)[10]. The blot was hybridized with a full-length octopine synthase (OCS) terminator sequence as probe, which was excised from pART7[31] with *Bam*HI and *Not*I digestion, gel purified, and radioactively labeled with [α-$^{32}$P] dCTP using the DecaLabel DNA Labeling Kit (Thermo Fisher Scientific Cat No. K0622) according to the manufacturer's instructions. The labeled DNA probe was purified using G-25 columns (Bio-strategy Cat No. 27-5325-01).

For detection of *GUS* and *EIN2* siRNAs, 20 μg of total RNA samples were separated in 17% denaturing acrylamide gel (19:1 acrylamide:bis Ambion Cat No. AM9022), electroblotted and UV crosslinked to HyBond-N$^+$ membrane (GE Healthcare), and hybridized with 200-nt *GUS* and *EIN2* sense RNA probe[10].

**Sodium periodate treatment of RNA (β-elimination)**. Treatment of RNA with sodium periodate (NaIO4) (Sigma Aldrich Cat No.71859) was performed according to Ebhardt et al.[36]. In brief, radiolabeled sRNA (0.025 pmol. mixed with 2 μg of total RNA from Col-0) or total RNA of Col-0 or T2 hpEIN2 plants (10 μg) was incubated in 15 μL of 10 mM HEPES (Sigma Aldrich Cat No. H3375), pH 7.0 and 100 mM sodium periodate at 22 °C for 10 min. Following this, 15 μL of formamide loading dye (supplemented with 5 mM EDTA) was then added, and the mixture was heated in boiling water for 30 min before loading.

**Preparation of differentially phosphorylated sRNA**. T7 RNA polymerase transcription of 5′ mono-, di- and tri-phosphorylated 24-nt sRNAs was performed using guanosine monophosphate, guanosine diphosphate, or GTP[18]. Sequences of the DNA oligonucleotides containing T7 promoter and *EIN2* sequences are shown in Supplementary Data 1.

**Alkaline phosphatase treatment of sRNAs**. Total RNA (10~20 μg) was incubated for 2 h at 37 °C in 100 μL reaction containing 1× CutSmart Buffer (New England Biolabs) and a total of 140 units of Calf Intestinal alkaline phosphatase (CIP) (New England BioLabs Cat No. M0525) (50, 30, 30, and 30 units were added to the reaction at 0, 30, 60, and 90 min). After incubation, RNA was purified with phenol/chloroform extraction, precipitated with 10 μL 3 M NaOAc and 250 μL of ethanol at −20 °C overnight, and dissolved in 10 μL H$_2$O. sRNA northern

hybridization analysis of the CIP-treated RNA and untreated samples was performed the same way as described above.

**McrBC-digestion PCR and bisulphite PCR**. Plant genomic DNA (~500 ng) was digested with 30 units of McrBC (NEB Cat No. M0272) in a 50 µL reaction volume at 37 °C overnight. For McrBC-minus controls, the same amount of DNA was incubated overnight at 37 °C in 50 µL reaction volumes containing the same buffer, but without the McrBC enzyme. 1 µL (50 ng) of digested and undigested DNA of each sample was used to set up PCR reactions using Taq DNA polymerase (NEB Cat No. M0273) along with ThermoPol buffer (NEB). The PCR product was electrophoresed on a 2% agarose gel, stained with ethidium bromide, and visualized by UV illumination.

Bisulfite conversion and purification were performed using the EpiTect Bisulfite kit (QIAGEN Cat No. 59124) following the procedures recommended by the manufacturer. Bisulfite PCR was performed as a nested PCR (two PCR reactions). The primers used in the first and second round PCR was listed in the Supplementary Data 1. The PCR cycles were as follows[37]: 12 min at 94℃ followed by 10 cycles of 1 min at 94℃, 2:30 min at 50℃, 1:30 min at 72℃, and 30 cycles with 1 min at 94℃, 1:30 min at 55℃, 1:30 min at 72℃, with a final extension of 10 min at 72℃. The PCR products from the second PCR were purified using QIAquick PCR purification kit (Qiagen Cat No. 28104) following the manufacturer's instructions. Approximately 50–200 ng of purified bisulfite PCR product was sequenced with BigDye Terminator V3.1 premix (Applied Biosystems) using one of the nested primers. Cytosine methylation levels were determined using the following procedure[38]: trace file data of the sequenced PCR products were opened using the BioEdit software (https://bioedit.software.informer.com), exported to Microsoft Excel using the 'Export trace values (tab-delimited text)' feature, and the relative peak heights of cytosines and thymines calculated to indicate the relative degree of methylation at each cytosine location.

**Analysis of small RNA sequencing data**. Cutadapt version 1.12 (https://cutadapt.readthedocs.io/en/stable/installation.html) was used to trim the adaptor sequences and filter out >35 nt or <18 nt sequences. The clean reads were mapped to reference hpEIN2 and hpGUS sequences, without mismatch, using Bowtie version 1.2.3 (http://bowtie-bio.sourceforge.net/index.shtml). sRNA reads were normalized against total reads including those mapped to the transgenes and Nicotiana or Arabidopsis genomes.

**Reporting summary**. Further information on research design is available in the Nature Research Reporting Summary linked to this article.

## Data availability
The small RNA sequencing data is accessible via GSE178565. Source data are provided with this paper.

## Code availability
Code used in small RNA sequencing analysis is available at GitHub [https://github.com/CSIRO-RNA/size_distribution-of-sRNAs/blob/main/Size_distribution.pl].

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

## Acknowledgements

We thank Carl Davies for photography and Craig Wood for supporting sRNA sequencing of RdDM mutant samples. We also thank Pablo Vera for the *ago4-2* mutant (*ocp11*), Peter Unrau for helpful discussions and the three reviewers for constructive comments. D.Z. was supported by a China Scholarship Council (CSC) scholarship and University of Wollongong Ph.D. tuition waiver, and C.Z. was supported by a CSC scholarship.

## Author contributions

D.Z. performed the majority of experiments; C.Z. conducted bioinformatic analysis of sRNA deep sequencing data; N.S. performed bisulfite sequencing and data analysis and some northern blot hybridization; R.D. contributed to data analysis and writing; I.G. contributed to the RdDM mutant experiment; S.S had input in project development and writing; R.Z. contributed to project initiation, supervision of D.Z. and writing; M.B.W. initiated and designed the work, performed some of the sRNA northern analysis, and did most of the writing.

## Competing interests

The authors declare no competing interests.
