## [Peer Review File · Nature Communications]

Nucleotide mismatches prevent intrinsic self-silencing of hpRNA transgenes to enhance RNAi stability in plantsReviewers' Comments:

Reviewer #1:

Remarks to the Author:

This manuscript showed that imperfect inverted repeat construct due to the C to T substitutions in the stem region induced an efficient silencing when compared with the silencing efficiency by the perfect match inverted repeat construct. In addition, the authors showed that the siRNAs originated from such imperfect IR construct possibly suffered an additional phosphorylation. The efficient silencing by imperfect IR construct was associated with the reduced cytosine methylation in the flanking promoter region.

The silencing using imperfect IR construct has been reported (ref 27 in this manuscript). So the C to T substitution in the stem region itself was not novel. Although the authors showed the improved silencing efficiency by the imperfect IR construct, we can obtain the strong silencing events by using conventional perfect IR construct.

Next, the authors showed that the improved silencing efficiency by imperfect IR construct was due to the decreased DNA methylation in the flanking promoter region. The bisulfate sequencing clearly showed the reduced DNA methylation. However, strong RNA silencing by perfect IR construct (ex. hpGUS[WT] event 6 (Fig. 4) have the strongly methylated promoter region. Other inconsistency between DNA methylation and silencing efficiency can be observed in several transgenic events (ex. hpEIN2[G:U] event 17 (strong silencing in Fig. S5a, and strong DNA methylation in Fig. 5a). DNA methylation will be affected by the insertion locus in the genome. So the relationship between the DNA methylation in the promoter region and silencing efficiency should be carefully discussed.

In page 12 first paragraph, the authors suggest as follows: "strong DNA methylation inside the perfect inverted-repeat DNA, as well as its spread to the upstream promoter, was not affected in the RdDM mutants". In general, both NRPD1 (RNA polIV) and NRPE1 (RNA polV) are involved in the RdDM and these two factors differentially act on the DNA methylation. Therefore, the above conclusion should be mentioned after the authors provide more detailed and focused studies using both mutants for plant-specific RNA polymerases and also other DNA methylation factors.

Finally, the authors showed the small differences in the gel mobilities of the siRNAs from the perfect IR and imperfect IR constructs. As the authors mentioned, the diphosphorylated or more phosphorylated siRNAs cannot be sequenced by NGS. So, the NGS data in the Fig. 8 showed the normally processed siRNA profile not for additionally phosphorylated siRNAs. However, the authors mentioned in page 13 that the main sizes of siRNAs by NGS were the same between the siRNAs from perfect IR and imperfect IR constructs, and also that therefore the siRNAs from imperfect IR constructs should be subjected to the chemical modification such as phosphorylation. This logic would totally mislead the readers. The data presented by Fig. 7 should be re-tried with more high resolution using sequencing gels, because the miR168 also showed some mobility shift after dephosphorylation treatment, because the authors said that miR168 resembled perfect IR construct-derived siRNAs (page 14, line 318). So miR168 serves as a control.

These points mentioned above made me against the publication of this manuscript in the Nature Communication. I recommend the authors should submit this to more specialized journal after the appropriate revision.

minor points.

- page 13, line 288-290. The explanation for secondary siRNA formation. The data in the Figure S11 showed the proportion of secondary GUS siRNAs were not so high. So, the discussion in this part should be revised.

Reviewer #2:

Remarks to the Author:

The authors have discovered a very interesting and significant phenomenon that G:U hpRNA transgenes show less DNA methylation and self-silencing than traditional hpRNA transgenes, and that

they are more likely to be stably expressed over generations. The research has also provided some insights into the mechanisms. This research is definitely suitable for publication in Nature Communications, however, it is quite a complex story and I recommend that the manuscript and figures be revised accordingly to my comments below. In particular, the DNA methylation data need to be clarified and Figure 6 and S7 need to include non-transgenic Col-0, ago4-2 and nrpd1a-3 as controls to confirm that hairpins are responsible for levels of DNA methylation in the endogenous target and flanking EIN2 and PDS sequences.

Line 53: "The rasiRNAs, however, are unique to plants and function to direct de novo cytosine methylation at the cognate DNA, a transcriptional gene silencing mechanism known as RNA-directed DNA methylation (RdDM)." RNA-directed chromatin modification also occurs in fungi and animals, most likely by a similar mechanism.

Line 94: Suggest you mention "Nicotiana tabacum lines PPGH11 and PPGH24 expressing the β -glucuronidase (GUS) reporter gene as the RNAi target" to make it clear what the RNAi assay involves.

Figure 1: hpRNA constructs should be in the same order in Figs 1A, B and C (and other figures). What are the error bars in Fig. 1B? This info should be in legend. Also need standard error bars based on n = biological replicates (i.e. independent lines) in Figure 1C.

Line 107: Suggest changing "untransformed" to "PPGH11 and PPGH24 GUS-expressing control lines"

Line 111: Looks like 72 (not 71) of the 74 hpGUS[G:U] lines (95.9%) tested showed strong RNAi? Also cite Fig. 1B here. In general, Figure panels need to be cited more in the text.

Figure 2. Suggest adding an additional panel (i.e. a histogram) to Figure 2 based on the data in Fig. 2C for the two hpGUS [WT] and three hpGUS [G:U] lines showing mean hypocotyl length \pm SE (based on n = 20 biological replicates, i.e. 20 independent T2 lines) along with statistical analysis

Figure 4:

- In the Figure 4B legend, need to mention that the arrows represent the binding site of PCR primers used in McrBC digestion PCR. Only two hpGUS[WT] lines were used for the bisulphite sequencing but is probably enough given the experiment was repeated for hpEIN2 with three replicates.

- Why is the hpGUS [1:4] methylation data included in Figure 4? It is not included in the hpEIN2 experiments (Figure 5) and should be removed from Figure 4 (but kept it in Figure 1) – including the hpGUS [1:4] data is an unnecessary complication and diversion for the reader.

- Suggest the authors add an additional panel (i.e. histogram) to Figure 4 based on the data in Fig. 4C for the two hpGUS [WT] and three hpGUS [G:U] lines showing mean 35S methylation for CG, CHG and CHH methylation \pm SE (based on n = 2 or 3 biological replicates, i.e. independent lines) across the 239 nt of BS sequence in common between the two constructs, along with statistical analysis. Furthermore, in Figure 4C, I suggest removing the extra 193 nucleotides of upstream 35S BS sequence data for hpGUS[G:U] lines because the authors don't have the hpGUS[WT] control data for this upstream 35S sequence, and also because it would allow precise alignment of the 35S nucleotides between the hpGUS [WT] and hpGUS [G:U] lines in the figure. (Currently, not even the hpGUS [WT] and hpGUS [1:4] 35S nucleotides align exactly in Figure 4C, but as mentioned above, I suggest removing the hpGUS [1:4] data from Figure 4 anyway.) In the two hpGUS [WT] lines #6 and #7, there is a high level of CHG methylation of three nucleotides at the 3' end of the BS sequence data in Figure 4C and the reverse primer for the BS sequencing seems to be in the GUS sequence – do these nucleotides correspond to hpGUS [WT] sequence included in the BS sequence data in Figure 4C for the two hpGUS [WT] lines? It should be clear in the figure and clearly stated in the legend that no hpGUS sequences are included Figure 4C.

Line 192-194. The data in Figures 5A and S5 don't support the statement: "The level of DNA methylation, as judged by reduced PCR amplification of McrBC-digested DNA, showed a good inverse correlation with the extent of EIN2 RNAi (Figure S5A)."

Line 199, Figure 5B not 5C

Figure 5. In the legend for Figure 5A, need to clearly define that the arrows represent the binding site of PCR primers used in McrBC digestion PCR. Figure 5 legend:

- Line 707, Figure 2 and Figure S5?

- Line 709, change "or" to "and"

- The entire legend for Figure 5C-E needs to be written more clearly and well organized

- What are the error bars in Figure 5D and E, and are they based on technical replicates for the qPCR or biological replicates (independent DNA extractions followed by qPCR). The number of independent DNA extractions (biological replicates) used needs to be defined in the legend of Figure 5D and E. Also statistical analysis based on biological replicates needs to be added to Figure 5D and E.

As for Figure 4, I suggest the authors add an additional panel to Figure 5 showing mean 35S methylation for CG, CHG and CHH \pm SE (based on n = 3 biological replicates, i.e. independent lines), along with statistical analysis. Furthermore, a cartoon of the hpPDS construct needs to be added to Figure 5C showing the location of the primers used for the hpPDS McrBC PCR and qPCR.

Figure 5B and Figure 6: the bisulfite sequencing data is not clearly presented and clearly explained in the legends. One could assume that the bisulfite sequencing data assays methylation of Cs in a 5'-3' direction on the sense coding sequence (top strand) of the first arm of the various hairpin sequences. However, there are no Cs on the upper coding strand of DNA construct for the hpRNA(G:U) constructs. Therefore, the bisulfite data must correspond to assaying methylation of Cs in a 3'-5' direction on the bottom non-coding sequence of the first arm of the hpRNA(G:U) constructs? Or is it combined for the top (5' to 3') and bottom (3' to 5') strands of the DNA? This should be made clear in the Figures and in the legends. In Figure 6, the representation of the bisulphite data for the endogenous target EIN2 sequences also needs to be clarified; does it represent methylation on the upper coding strand of DNA in Figure 6 or methylation on both DNA strands of the endogenous EIN2 locus?

It could be very interesting to assess methylation of Cs on both the upper and lower DNA strands endogenous target EIN2 sequences; one would expect the CG and CHG methylation patterns to be the same on both strands due to maintenance methylation following cell division. However, for CHH methylation, in view of the large amounts of sense G:U siRNA and low amount of antisense WT (G:C) siRNAs produced from the hpRNA (G:U) (Figure 8), could there be a strand-specific effect on CHH methylation or the endogenous target EIN2 sequences? If so, a strand-specific effect of CHH methylation should also be observed on the second arm of the inverted repeat in the hpRNA (G:U) lines. This data could address whether or not there is a DNA strand-specific aspect to RdDM. These comments also apply to Figure S7 (hpPDS).

As for Figure 4 and 5, I suggest the authors add an additional panel to Figures 6 and S7 showing mean 35S and target gene methylation for CG, CHG and CHH \pm SE (based on n = number of biological replicates, i.e. independent lines), along with statistical analysis, for both the top and bottom strands of DNA.

Figure 6 and S7 need to include bisulfite sequencing of non-transgenic Col-0, ago4-2 and nrpd1a-3 as controls to confirm that hairpins are responsible for levels of DNA methylation in the endogenous target and flanking EIN2 and PDS sequences shown in the Figures.

Line 205-207, need to cite Figures 2, 5A and S5A in this sentence.

Line 213, change (Figure 5D, E, F) to (Figure 5C, D, E)

Line 215, add....based on McrBC qPCR (Figure 5E).

Line 219, change "result" to "results"

Line 230, nrpd1a-3 is a SALK T-DNA mutant and SALK T-DNAs carry a 35S promoter that has been shown to interfere with, and potentially cause transcriptional down regulation of any other 35S promoters introduced into the SALK mutants (Daxinger et al. 2008 TIPS 13, 4). This could explain the high methylation of the 35S promoter in nrpd1a/hp[G:U]-13 in Figure 6. Furthermore, the enhanced RNAi observed with the hpRNA (WT) in the nrpd1a-3 mutant compared to the hpRNA (WT) in WT could be due to transcriptional down-regulation of the 35S:hpRNA transgene by the SALK T-DNA 35S promoter rather than the siRNAs derived from the hpRNA, which could result in sufficient siRNAs being produced from the hpRNA for near complete RNAi but insufficient siRNAs being produced from the hpRNA to completely silence the 35S:hpRNA transgene in nrpd1a lines. It is also unclear whether the enhanced RNAi observed with the hpRNA (G:U) compared to hpRNA (G:U) in the nrpd1a-3 mutant could be due in part to the transcriptional down-regulation of the 35S:hpRNA (G:U) by the SALK T-DNA 35S promoter rather than siRNAs derived from the hpRNA. However, this potential problem with interpreting the data for nrpd1a-3 is not a problem with the ago4-2 (ocp11) mutant as it is an EMS-induced mutant (not a SALK T-DNA mutant).

Line 232, the ocp11 mutant should be called ago4-2 (not ocp11) (as suggested by Agorio and Vera, 2007, who isolated the mutant). Also, this mutant has only been demonstrated to be incompletely dominant for the GUS reporter system that was used to recover the ago4-2 (not ocp11) mutant (Agorio and Vera, 2007). Therefore, in the context of ago4 mutants in general, DNA methylation and this manuscript, ago4-2 shouldn't be referred to as a dominant-negative mutant.

Line 247 to 250, the lack of nrpd1a hpPDS[WT] lines developing strong photo-bleaching may not indicate that the loss of Pol IV reduced the extent of PDS RNAi; this reduced extent of PDS RNAi in nrpd1a-3 may be due to transcriptional down-regulation of the 35S:hpPDS transgene by the SALK T-DNA 35S promoter as mentioned above.

Line 254, see comment above regarding nrpd1a/hp[G:U]-13; SALK T-DNAs potentially causing transcriptional down regulation of any other 35S promoters introduced into the SALK mutants.

Line 259-261, is important to be clear whether the DNA methylation data in Figure 6 is for both the top and bottom strands of DNA in the endogenous EIN2 sequences; as mentioned above suggest an additional panel to Figures 6 (and S7) showing mean 35S and target gene methylation for CG, CHG and CHH \pm SE (based on n = number of biological replicates, i.e. independent lines), along with statistical analysis, for both the top and bottom strands of DNA. The CHH methylation is most different between the EIN2

Line 261 to 267, the lower level of EIN2 target methylation in nrpd1a hpPDS[WT] lines may not indicate that Pol IV is involved in target gene methylation; as indicated earlier, this reduced target gene methylation in nrpd1a-3 may be due to transcriptional down-regulation of the 35S:hpEIN2 transgene by the SALK T-DNA 35S promoter and therefore less EIN2 primary siRNAs derived from the hairpin. Thus, it may be that all hairpins tested induce methylation of endogenous target sequences independent of Pol IV and is simply depends on the abundance of primary siRNAs derived from the hairpin. Thus, both WT and G:U hpRNA transgenes may induced RdDM independently of Pol IV.

Line 310, could the under-representation of hpEIN2[G:U]-derived antisense sRNAs compared to hpEIN2[WT]-derived antisense siRNAs in the sRNA-seq data also be due to AGO1's preferential loading siRNA duplexes with 5'U on what will be the guide sRNA?

Line 320, change "different" to "variable"

Line 409-412: The differential gel mobility and 5' phosphorylation of both 21 and 24 WT and G:U siRNAs suggests that the variable 5' phosphorylation isn't due to DCL1 versus DCL4 processing, as DCL4 dependent 24 nt siRNAs are also different between WT and G:U siRNAs.

Fig S9 Why include the hpGUS [1:4]?

Line 692, need to clearly define that the arrows represent the binding site of PCR primers used in McrBC digestion PCR.

Line 705-706, need to define that the arrows represent the binding site of PCR primers used in McrBC digestion PCR.

Figure S5 What are the error bars in Figure S 5D and E, and how many biological replicates are they based on? This information should be in the legend.

The legends in general should be revised according to the comments above on several of the legends.

There is some inconsistent formatting of references.

Reviewer #3:

Remarks to the Author:

The Zhang et al. study focuses on developing an understanding of the variability and modulation effects that are common in gene suppression studies that induce RNAi by hairpin-containing transgenes. There are important fundamental and practical implications to this question, particularly with regard to RdDM pathway mechanisms and the future design of more effective and stable constructs for targeted gene suppression.

The investigators make a compelling argument that perfect hairpins serve as targets for cytosine methylation, which serves to suppress transgene expression and RNAi effectiveness. This is not new information but their model systems are very effective for demonstrating the effect at high resolution. Inclusion of three target loci to the study is valuable, and selection of PDS as one of the targets provides the novel and unexpected observation of developmental modulation of RNAi occurring from cotyledon to true leaf stages. This observation is quite intriguing. The incorporation of various levels of nucleotide substitution for hairpin mismatch provides evidence that quantitative differences are attributable to the hairpin composition and its associated effect on local (promoter) methylation. An additional important test of the hypothesis was the inclusion of RdDM mutants to demonstrate the influence of this pathway in the methylation changes observed.

It appears that the weaknesses in the study lie in the rather substantial speculation that is included in the Results. Evidence suggests that the RdDM pathway, as defined by the particular mutants tested here, does not regulate the DNA methylation observed within the IR structure of the hpRNA transgene. But this observation leaves open a number of unaddressed possibilities such as noncanonical features of RNA-dependent DNA methylation yet undefined, an alternative system for site-specific DNA methylation, or features of the genes/mutants/experimental design selected for this study that do that allow for proper detection of site-specific methylation onset, perhaps, at the point of floral dip, which may no longer be subject to RdDM-mediated reinforcement. To leave the argument as "intrinsic methylation...is not affected in RdDM mutants" surmises negative data without understanding but merely speculating. Similarly, differential processing and diphosphorylation or polyphosphorylation of siRNAs is deduced from gel migration assays alone, with relatively little to conclude beyond the observation that mismatched hairpins condition distinct outcomes. These observations are again supported largely with additional speculation. It is not clear that the study provides meaningful insight into alternative RdDM processes and/or siRNA processing. For this reason, I question whether there is sufficient meaningful interpretation of the system to yet merit publication in Nat Comm, although the data presented here are likely to significantly impact the technical implementation of RNAi in gene suppression.

Response to reviewers' comments

Reviewer #1 (Remarks to the Author):

This manuscript showed that imperfect inverted repeat construct due to the C to T substitutions in the stem region induced an efficient silencing when compared with the silencing efficiency by the perfect match inverted repeat construct. In addition, the authors showed that the siRNAs originated from such imperfect IR construct possibly suffered an additional phosphorylation. The efficient silencing by imperfect IR construct was associated with the reduced cytosine methylation in the flanking promoter region.

General response: We appreciate the specific comments by the reviewer, which we have addressed as detailed below including new experiments, which we believe has helped to improve the paper significantly.

The silencing using imperfect IR construct has been reported (ref 27 in this manuscript). So the C to T substitution in the stem region itself was not novel. Although the authors showed the improved silencing efficiency by the imperfect IR construct, we can obtain the strong silencing events by using conventional perfect IR construct.

Response: Whilst we agree that incorporating G:U base-pairing in RNAi design was shown by cited reference #27, where the authors' focus was aimed at increasing the IR DNA stability in bacteria to facilitate cloning, no comparative in depth molecular analysis using G:U base pairing to ameliorate self-induced transcriptional silencing of traditional hpRNA transgenes in plants has been reported. This study confirms the long-suspected issues confounding IR DNA self-silencing, and more importantly, has made discoveries that introducing nucleotide mismatches to the sense strand of IR DNA (including C to T changes giving rise to G:U base-pairing), can effectively prevent self-silencing of hpRNA and enhance uniformity of RNAi. These insights and novel applications of nucleotide mismatches were not reported in Ref #27 or any other previous studies.

Next, the authors showed that the improved silencing efficiency by imperfect IR construct was due to the decreased DNA methylation in the flanking promoter region. The bisulfate sequencing clearly showed the reduced DNA methylation. However, strong RNA silencing by perfect IR construct (ex. hpGUS[WT] event 6 (Fig. 4) have the strongly methylated promoter region. Other inconsistency between DNA methylation and silencing efficiency can be observed in several transgenic events (ex. hpEIN2[G:U] event 17 (strong silencing in Fig. S5a, and strong DNA methylation in Fig. 5a). DNA methylation will be affected by the insertion locus in the genome. So the relationship between the DNA methylation in the promoter region and silencing efficiency should be carefully discussed.

Response: Thank you for the comments. Careful considerations were given accordingly when revising the text of the paper. One of the points we present is that even the relatively strong RNAi lines of the traditional hpRNA construct have moderate levels of DNA methylation in the 35S promoter, such as hpGUS[WT] event 6 (Figure 4). We think that this "moderate" promoter methylation has not reached the level required to completely shut down the promoter but nevertheless compromise the transcriptional activity, preventing the full potential of RNAi by the traditional hpRNA constructs. We did not perform bisulfite sequencing on the lines of low or no RNAi, but MCR-PCR does suggest more intense 35S promoter methylation in these lines. We agree that transgene insertions play an important role in the promoter methylation of the hpRNA

transgenic population, which is supported by the result with the RdDM mutants: RNAi induced by the hpRNA transgenes becomes more uniform in the mutants indicating the existence of insertion-dependent promoter methylation that is released in the RdDM mutants, but GU hpRNA transgenes still induce stronger RNAi than the traditional hpRNA transgenes indicating the existence of IR-induced intrinsic DNA methylation of the traditional hpRNA transgenes that is not affected in the RdDM mutants.

In page 12 first paragraph, the authors suggest as follows: “strong DNA methylation inside the perfect inverted-repeat DNA, as well as its spread to the upstream promoter, was not affected in the RdDM mutants”. In general, both NRPD1 (RNA polIV) and NRPE1 (RNA polV) are involved in the RdDM and these two factors differentially act on the DNA methylation. Therefore, the above conclusion should be mentioned after the authors provide more detailed and focused studies using both mutants for plant-specific RNA polymerases and also other DNA methylation factors.

Response: To address this comment, we have revised the texts by emphasizing specifically the RdDM components AGO4 and POL IV, rather than the whole RdDM pathway, in the methylation of IR DNA of the hpRNA transgene. Pol IV is the dominant upstream RdDM factor in 24nt siRNA biogenesis whereas AGO4 is the core downstream effector of the canonical RdDM, and the changes of methylation in target gene but not in the IR transgene DNA in these mutants do point to a different involvement of RdDM in IR methylation. We agree that other RdDM/methylation factors including histone methylation factors could also be involved in IR methylation, but it would take a separate, more focused study to examine these other factors.

Finally, the authors showed the small differences in the gel mobilities of the siRNAs from the perfect IR and imperfect IR constructs. As the authors mentioned, the diphosphorylated or more phosphorylated siRNAs cannot be sequenced by NGS. So, the NGS data in the Fig. 8 showed the normally processed siRNA profile not for additionally phosphorylated siRNAs. However, the authors mentioned in page 13 that the main sizes of siRNAs by NGS were the same between the siRNAs from perfect IR and imperfect IR constructs, and also that therefore the siRNAs from imperfect IR constructs should be subjected to the chemical modification such as phosphorylation. This logic would totally mislead the readers. The data presented by Fig. 7 should be re-tried with more high resolution using sequencing gels, because the miR168 also showed some mobility shift after dephosphorylation treatment, because the authors said that miR168 resembled perfect IR construct-derived siRNAs (page 14, line 318). So miR168 serves as a control.

Response: We thank the reviewer for these comments. We performed additional experiments, including i) beta-elimination assay to rule out any differences in 3' methylation of the siRNAs (revised Figure 7C), ii) running a longer denaturing PAGE gel to provide better separation of phosphatase-treated and untreated siRNAs (revised Figure 7D), and iii) preparing mono-, di-, and tri-phosphorylated siRNAs as gel markers (revised Figure 7E). We also made significant modifications of the texts accordingly.

minor points.

- page 13, line 288-290. The explanation for secondary siRNA formation. The data in the Figure S11 showed the proportion of secondary GUS siRNAs were not so high. So, the discussion in this part should be revised.

Response: We have revised the discussion accordingly.

Reviewer #2 (Remarks to the Author):

The authors have discovered a very interesting and significant phenomenon that G:U hpRNA transgenes show less DNA methylation and self-silencing than traditional hpRNA transgenes, and that they are more likely to be stably expressed over generations. The research has also provided some insights into the mechanisms. This research is definitely suitable for publication in Nature Communications, however, it is quite a complex story and I recommend that the manuscript and figures be revised accordingly to my comments below. In particular, the DNA methylation data need to be clarified and Figure 6 and S7 need to include non-transgenic Col-0, ago4-2 and nrpd1a-3 as controls to confirm that hairpins are responsible for levels of DNA methylation in the endogenous target and flanking EIN2 and PDS sequences.

General response: We appreciate the positive assessment and the many constructive comments on our manuscript. We have addressed these comments as detailed below, including new experiments that we think have helped to improve the paper significantly.

Line 53: “The rasiRNAs, however, are unique to plants and function to direct de novo cytosine methylation at the cognate DNA, a transcriptional gene silencing mechanism known as RNA-directed DNA methylation (RdDM).” RNA-directed chromatin modification also occurs in fungi and animals, most likely by a similar mechanism.

Response: We modified the sentence by removing “are unique to plants”.

Line 94: Suggest you mention “Nicotiana tabacum lines PPGH11 and PPGH24 expressing the β -glucuronidase (GUS) reporter gene as the RNAi target” to make it clear what the RNAi assay involves.

Response: Change is made accordingly.

Figure 1: hpRNA constructs should be in the same order in Figs 1A, B and C (and other figures). What are the error bars in Fig. 1B? This info should be in legend. Also need standard error bars based on n = biological replicates (i.e. independent lines) in Figure 1C.

Response: Changes are made accordingly in figure and legends

Line 107: Suggest changing “untransformed” to “PPGH11 and PPGH24 GUS-expressing control lines”

Response: Change is made accordingly.

Line 111: Looks like 72 (not 71) of the 74 hpGUS[G:U] lines (95.9%) tested showed strong RNAi? Also cite Fig. 1B here. In general, Figure panels need to be cited more in the text.

Response: The error is corrected and Figure is cited.

Figure 2. Suggest adding an additional panel (i.e. a histogram) to Figure 2 based on the data in Fig. 2C

for the two hpGUS [WT] and three hpGUS [G:U] lines showing mean hypocotyl length \pm SE (based on n = 20 biological replicates, i.e. 20 independent T2 lines) along with statistical analysis

Response: We revised Figure 2 by adding a boxplot showing the distribution of hypocotyl length. The two populations consisted of independent lines, and we think boxplot is a better way to show the difference in hypocotyl length.

Figure 4:

- In the Figure 4B legend, need to mention that the arrows represent the binding site of PCR primers used in McrBC digestion PCR. Only two hpGUS[WT] lines were used for the bisulphite sequencing but is probably enough given the experiment was repeated for hpEIN2 with three replicates.
- Why is the hpGUS [1:4] methylation data included in Figure 4? It is not included in the hpEIN2 experiments (Figure 5) and should be removed from Figure 4 (but kept it in Figure 1) – including the hpGUS [1:4] data is an unnecessary complication and diversion for the reader.
- Suggest the authors add an additional panel (i.e. histogram) to Figure 4 based on the data in Fig. 4C for the two hpGUS [WT] and three hpGUS [G:U] lines showing mean 35S methylation for CG, CHG and CHH methylation \pm SE (based on n = 2 or 3 biological replicates, i.e. independent lines) across the 239 nt of BS sequence in common between the two constructs, along with statistical analysis. Furthermore, in Figure 4C, I suggest removing the extra 193 nucleotides of upstream 35S BS sequence data for hpGUS[G:U] lines because the authors don't have the hpGUS[WT] control data for this upstream 35S sequence, and also because it would allow precise alignment of the 35S nucleotides between the hpGUS [WT] and hpGUS [G:U] lines in the figure. (Currently, not even the hpGUS [WT] and hpGUS [1:4] 35S nucleotides align exactly in Figure 4C, but as mentioned above, I suggest removing the hpGUS [1:4] data from Figure 4 anyway.) In the two hpGUS [WT] lines #6 and #7, there is a high level of CHG methylation of three nucleotides at the 3' end of the BS sequence data in Figure 4C and the reverse primer for the BS sequencing seems to be in the GUS sequence – do these nucleotides correspond to hpGUS [WT] sequence included in the BS sequence data in Figure 4C for the two hpGUS [WT] lines? It should be clear in the figure and clearly stated in the legend that no hpGUS sequences are included Figure 4C.

Response: We appreciate these comments regarding Figure 4. We revised the figure accordingly by i) removing the hpGUS[1:4] data, ii) adding another panel to show mean 35S promoter methylation values, iii) re-analyzing and re-drawing the bisulfite sequencing figure to align the hpGUS[WT] and hpGUS[G:U] sequence properly, and iv) modifying the figure legends.

Line 192-194. The data in Figures 5A and S5 don't support the statement: "The level of DNA methylation, as judged by reduced PCR amplification of McrBC-digested DNA, showed a good inverse correlation with the extent of EIN2 RNAi (Figure S5A)."

Response: The sentence was deleted.

Line 199, Figure 5B not 5C

Response: This is corrected

Figure 5. In the legend for Figure 5A, need to clearly define that the arrows represent the binding site of PCR primers used in McrBC digestion PCR. Figure 5 legend:

Response: Figure legends are modified.

- Line 707, Figure 2 and Figure S5?

Response: This is corrected.

- Line 709, change “or” to “and”

Response: Change is made.

- The entire legend for Figure 5C-E needs to be written more clearly and well organized
- What are the error bars in Figure 5D and E, and are they based on technical replicates for the qPCR or biological replicates (independent DNA extractions followed by qPCR). The number of independent DNA extractions (biological replicates) used needs to be defined in the legend of Figure 5D and E. Also statistical analysis based on biological replicates needs to be added to Figure 5D and E.

As for Figure 4, I suggest the authors add an additional panel to Figure 5 showing mean 35S methylation for CG, CHG and CHH \pm SE (based on $n = 3$ biological replicates, i.e. independent lines), along with statistical analysis. Furthermore, a cartoon of the hpPDS construct needs to be added to Figure 5C showing the location of the primers used for the hpPDS McrBC PCR and qPCR.

Response: We modified the Figure 5 legends accordingly to provide clearer explanations and added another panel to show the average DNA methylation levels in the promoter region, and a schematic diagram for the hpPDS construct showing the location of the analyzed regions.

Figure 5B and Figure 6: the bisulfite sequencing data is not clearly presented and clearly explained in the legends. One could assume that the bisulfite sequencing data assays methylation of Cs in a 5'-3' direction on the sense coding sequence (top strand) of the first arm of the various hairpin sequences. However, there are no Cs on the upper coding strand of DNA construct for the hpRNA(G:U) constructs. Therefore, the bisulfite data must correspond to assaying methylation of Cs in a 3'-5' direction on the bottom non-coding sequence of the first arm of the hpRNA(G:U) constructs? Or is it combined for the top (5' to 3') and bottom (3' to 5') strands of the DNA? This should be made clear in the Figures and in the legends. In Figure 6, the representation of the bisulphite data for the endogenous target EIN2 sequences also needs to be clarified; does it represent methylation on the upper coding strand of DNA in Figure 6 or methylation on both DNA strands of the endogenous EIN2 locus?

Response: Thank you for pointing this out. We modified the figure legends to explain that the top DNA strand was analyzed and presented.

It could be very interesting to assess methylation of Cs on both the upper and lower DNA strands endogenous target EIN2 sequences; one would expect the CG and CHG methylation patterns to be the same on both strands due to maintenance methylation following cell division. However, for CHH methylation, in view of the large amounts of sense G:U siRNA and low amount of antisense WT (G:C) siRNAs produced from the hpRNA (G:U) (Figure 8), could there be a strand-specific effect on CHH methylation or the endogenous target EIN2 sequences? If so, a strand-specific effect of CHH

methylation should also be observed on the second arm of the inverted repeat in the hpRNA (G:U) lines. This data could address whether or not there is a DNA strand-specific aspect to RdDM. These comments also apply to Figure S7 (hpPDS).

Response: We greatly appreciate this comment/suggestion. We performed bisulfite sequencing of the bottom strand of the *EIN2* genomic target in hpEIN2[WT] and hpEIN2[G:U] plant lines with relatively high amount of siRNAs and top-strand methylation in Col-0 and particularly *nrpd1a-3* where secondary siRNA production is minimized, and present the data in Figure S8 and Figure S7. The result is very interesting, suggesting that there is strand specificity in siRNA-directed DNA methylation, as suggested by the reviewer, with G:U modified sense siRNAs incapable of inducing methylation on the bottom genomic strand due to sequence mismatches. We are reluctant to draw a strong conclusion from this in the paper as we believe more detailed studies are needed to verify and understand this.

As for Figure 4 and 5, I suggest the authors add an additional panel to Figures 6 and S7 showing mean 35S and target gene methylation for CG, CHG and CHH \pm SE (based on n = number of biological replicates, i.e. independent lines), along with statistical analysis, for both the top and bottom strands of DNA.

Response: The additional panel is added.

Figure 6 and S7 need to include bisulfite sequencing of non-transgenic Col-0, ago4-2 and nrpd1a-3 as controls to confirm that hairpins are responsible for levels of DNA methylation in the endogenous target and flanking EIN2 and PDS sequences shown in the Figures.

Response: We performed the experiment on *EIN2* and added the data in Figure S8, which shows that the four CG sites on the 5' half of the 200 bp *EIN2* target sequence is already heavily methylated in the untransformed plants but the non-CG sites are largely unmethylated, which allows for the detection of hpRNA-induced non-CG methylation in the whole sequence but also CG methylation in the 3' half.

Line 205-207, need to cite Figures 2, 5A and S5A in this sentence.

Response: Changes are made.

Line 213, change (Figure 5D, E, F) to (Figure 5C, D, E)

Response: Change is made.

Line 215, add....based on McrBC qPCR (Figure 5E).

Response: Change is made.

Line 219, change "result" to "results"

Response: Change is made.

Line 230, nrpd1a-3 is a SALK T-DNA mutant and SALK T-DNAs carry a 35S promoter that has been shown to interfere with, and potentially cause transcriptional down regulation of any other 35S

promoters introduced into the SALK mutants (Daxinger et al. 2008 TIPS 13, 4). This could explain the high methylation of the 35S promoter in *nrpd1a*/hp[G:U]-13 in Figure 6. Furthermore, the enhanced RNAi observed with the hpRNA (WT) in the *nrpd1a-3* mutant compared to the hpRNA (WT) in WT could be due to transcriptional down-regulation of the 35S:hpRNA transgene by the SALK T-DNA 35S promoter rather than the siRNAs derived from the hpRNA, which could result in sufficient siRNAs being produced from the hpRNA for near complete RNAi but insufficient siRNAs being produced from the hpRNA to completely silence the 35S:hpRNA transgene in *nrpd1a* lines. It is also unclear whether the enhanced RNAi observed with the hpRNA (G:U) compared to hpRNA (G:U) in the *nrpd1a-3* mutant could be due in part to the transcriptional down-regulation of the 35S:hpRNA (G:U) by the SALK T-DNA 35S promoter rather than siRNAs derived from the hpRNA. However, this potential problem with interpreting the data for *nrpd1a-3* is not a problem with the *ago4-2* (*ocp11*) mutant as it is an EMS-induced mutant (not a SALK T-DNA mutant).

Response: This is a very interesting comment, and we have revised the discussion to address this (Page 15). We were aware of the potential TGS issue with the Salk T-DNA insertion mutants and were initially also concerned about the possible involvement of 35S promoter silencing in the weaker RNAi and other related observations of *nrpd1a*. However, the almost comparable levels of dsRNA intermediates and siRNAs detected in the 3 *nrpd1a*/hpEIN2[G:U] lines in comparison to the Col-0 lines (Figure 7B) made us believe that the T-DNA insertion has in general not caused strong TGS to the 35S promoter. However, further work is needed to clarify this, such as using an EMS mutant.

Line 232, the *ocp11* mutant should be called *ago4-2* (not *ocp11*) (as suggested by Agorio and Vera, 2007, who isolated the mutant). Also, this mutant has only been demonstrated to be incompletely dominant for the GUS reporter system that was used to recover the *ago4-2* (not *ocp11*) mutant (Agorio and Vera, 2007). Therefore, in the context of *ago4* mutants in general, DNA methylation and this manuscript, *ago4-2* shouldn't be referred to as a dominant-negative mutant.

Response: We changed "*ocp11*" to "*ago4-2*", and deleted "dominant-negative".

Line 247 to 250, the lack of *nrpd1a* hpPDS[WT] lines developing strong photo-bleaching may not indicate that the loss of Pol IV reduced the extent of PDS RNAi; this reduced extent of PDS RNAi in *nrpd1a-3* may be due to transcriptional down-regulation of the 35S:hpPDS transgene by the SALK T-DNA 35S promoter as mentioned above.

Response: Please see the response above re line 230.

Line 254, see comment above regarding *nrpd1a*/hp[G:U]-13; SALK T-DNAs potentially causing transcriptional down regulation of any other 35S promoters introduced into the SALK mutants.

Response: Please also see our response above.

Line 259-261, is important to be clear whether the DNA methylation data in Figure 6 is for both the top and bottom strands of DNA in the endogenous EIN2 sequences; as mentioned above suggest an additional panel to Figures 6 (and S7) showing mean 35S and target gene methylation for CG, CHG and CHH \pm SE (based on n = number of biological replicates, i.e. independent lines), along with

statistical analysis, for both the top and bottom strands of DNA. The CHH methylation is most different between the EIN2

Response: Changes are made accordingly in Figure, Figure legends, and text.

Line 261 to 267, the lower level of EIN2 target methylation in *nrrpd1a* hpPDS[WT] lines may not indicate that Pol IV is involved in target gene methylation; as indicated earlier, this reduced target gene methylation in *nrrpd1a-3* may be due to transcriptional down-regulation of the 35S:hpEIN2 transgene by the SALK T-DNA 35S promoter and therefore less EIN2 primary siRNAs derived from the hairpin. Thus, it may be that all hairpins tested induce methylation of endogenous target sequences independent of Pol IV and is simply depends on the abundance of primary siRNAs derived from the hairpin. Thus, both WT and G:U hpRNA transgenes may induced RdDM independently of Pol IV.

Response: Please see our response above regarding the potential of T-DNA insertion to cause 35S promoter inactivation. While reduced siRNA amounts in the *nrrpd1a* background would certainly account for the reduced target gene methylation, the significant amounts of dsRNA intermediates and siRNAs in the *nrrpd1a*/hpEIN2[G:U] lines make it less likely that 35S promoter inactivation plays a major role in the siRNA accumulation in the *nrrpd1a*/hpEIN2[WT] plants.

Line 310, could the under-representation of hpEIN2[G:U]-derived antisense sRNAs compared to hpEIN2[WT]-derived antisense siRNAs in the sRNA-seq data also be due to AGO1's preferential loading siRNA duplexes with 5'U on what will be the guide sRNA?

Response: Thanks for the thought. It is possible, and we added this in the discussion "The antisense siRNAs in the hpEIN2[G:U] lines are less abundant than the sense, G:U modified siRNAs (Figure 8). It is possible that the sense siRNAs were relatively enriched for 5' U and therefore preferentially loaded to AGO1 resulting in higher abundance than the antisense siRNAs."

Line 320, change "different" to "variable"

Response: Change is made.

Line 409-412: The differential gel mobility and 5' phosphorylation of both 21 and 24 WT and G:U siRNAs suggests that the variable 5' phosphorylation isn't due to DCL1 versus DCL4 processing, as DCL4 dependent 24 nt siRNAs are also different between WT and G:U siRNAs.

Response: We have performed further experiments on siRNA terminal modification (Figure 7C, D, E) and revised the text significantly.

Fig S9 Why include the hpGUS [1:4]?

Response: We removed the data on hpGUS[1:4].

Line 692, need to clearly define that the arrows represent the binding site of PCR primers used in McrBC digestion PCR.

Line 705-706, need to define that the arrows represent the binding site of PCR primers used in McrBC digestion PCR.

Response: Figure legends are revised to provide explanations.

Figure S5 What are the error bars in Figure S 5D and E, and how many biological replicates are they based on? This information should be in the legend.

Response: The figure legends are modified (now Figure S4).

The legends in general should be revised according to the comments above on several of the legends.

Response: Thank you for the suggestion.

There is some inconsistent formatting of references.

Response: Thank you for pointing this out.

Reviewer #3 (Remarks to the Author):

The Zhang et al. study focuses on developing an understanding of the variability and modulation effects that are common in gene suppression studies that induce RNAi by hairpin-containing transgenes. There are important fundamental and practical implications to this question, particularly with regard to RdDM pathway mechanisms and the future design of more effective and stable constructs for targeted gene suppression.

G

The investigators make a compelling argument that perfect hairpins serve as targets for cytosine methylation, which serves to suppress transgene expression and RNAi effectiveness. This is not new information but their model systems are very effective for demonstrating the effect at high resolution. Inclusion of three target loci to the study is valuable, and selection of PDS as one of the targets provides the novel and unexpected observation of developmental modulation of RNAi occurring from cotyledon to true leaf stages. This observation is quite intriguing. The incorporation of various levels of nucleotide substitution for hairpin mismatch provides evidence that quantitative differences are attributable to the hairpin composition and its associated effect on local (promoter) methylation. An additional important test of the hypothesis was the inclusion of RdDM mutants to demonstrate the influence of this pathway in the methylation changes observed.

General response: We appreciate the positive assessment and the helpful comments and suggestions. We have made significant amounts of changes to the data and texts which we believe have improved the manuscript.

It appears that the weaknesses in the study lie in the rather substantial speculation that is included in the Results. Evidence suggests that the RdDM pathway, as defined by the particular mutants tested here, does not regulate the DNA methylation observed within the IR structure of the hpRNA transgene. But this observation leaves open a number of unaddressed possibilities such as noncanonical features of RNA-dependent DNA methylation yet undefined, an alternative system for site-specific DNA methylation, or features of the genes/mutants/experimental design selected for this study that do that allow for proper detection of site-specific methylation onset, perhaps, at the point of floral dip, which may no longer be subject to RdDM-mediated reinforcement. To leave the

argument as “intrinsic methylation...is not affected in RdDM mutants” surmises negative data without understanding but merely speculating.

Response: To address these comments, we have revised the texts by emphasizing specifically the RdDM components AGO4 and POL IV, rather than the whole RdDM pathway, in the methylation of IR DNA of the hpRNA transgene. Pol IV is the dominant upstream RdDM factor in 24nt siRNA biogenesis whereas AGO4 is the core downstream effector of the canonical RdDM, and the result is clear that methylation in the IR transgene DNA is not affected in these two mutants whereas that of the genomic target is affected. This result does point to a different involvement of canonical RdDM in IR methylation. We agree that other RdDM/methylation factors including histone methylation factors could also be involved in IR methylation, but it would take a separate, more focused study to examine the other factors.

Similarly, differential processing and diphosphorylation or polyphosphorylation of siRNAs is deduced from gel migration assays alone, with relatively little to conclude beyond the observation that mismatched hairpins condition distinct outcomes. These observations are again supported largely with additional speculation. It is not clear that the study provides meaningful insight into alternative RdDM processes and/or siRNA processing. For this reason, I question whether there is sufficient meaningful interpretation of the system to yet merit publication in Nat Comm, although the data presented here are likely to significantly impact the technical implementation of RNAi in gene suppression.

Response: In our study we focus on the self-induced transcriptional silencing of the traditional hpRNA transgene design. We confirmed the long suspected intrinsic DNA methylation and self-silencing in the traditional hpRNA transgenes. While we agree that more work is needed to fully understand the factors involved in the self silencing, we provided clear demonstration that introducing nucleotide mismatches to the sense strand, disrupting the perfect IR DNA structure, including C to T changes (resulting in G:U base-pairing), can prevent the self-silencing and enhance uniformity of RNAi. These insights and novel applications of nucleotide mismatches are not previously reported and in our view are significant advancement of our understanding on IR DNA-associated silencing.

As further improvements on the part of siRNA modification, we performed additional northern hybridization experiments to characterize the siRNAs, i) showing that siRNAs from both traditional and G:U modified siRNAs are commonly methylated at the 3' hydroxyl, ii) running a longer gel to separate the two different siRNA populations before and after phosphatase treatments, and iii) including mono-, di- and tri-phosphorylated small RNAs as gel electrophoresis markers. We hope that these changes make the MS not only valuable technically but also broadly to the understanding of plant gene silencing. We therefore think that overall this work is worthy of publication in *Nature Communications*.

Reviewers' Comments:

Reviewer #1:

Remarks to the Author:

This revised manuscript was much improved. However, several points still remain to be refined.

Line 160-161: I cannot understand the differences between the following two phenomena; i.e., reduction in self silencing or increase of transcriptional stability.

Line 169: I do not like the "direct link between promoter methylation and reduced RNAi". As authors showed, 5 lines (#2,5,8,9,10) showed reduced PCR amplification after McrBC treatment. However, #6 line showed similar reduction in PCR amplification when compared with the #5 line's data. The #6 line showed strong RNAi phenotype, even though this line showed moderate methylation as shown in Fig. 4C,4E. Overall, the degree of DNA methylation and reduction of RNAi show a good correlation as mentioned by authors, but I wonder if the degree of RNAi is determined only by the DNA methylation in this case.

Line 398-400: It is likely that the large-sized dsRNA intermediates would be explained by poor Dicer processing even though the authors denied such possibility. I think that in the hpEIN2[WT] plants the transcription of inverted repeat sequences itself was suppressed by DNA methylation as indicated in this paper. So, the transcription rates of introduced inverted repeat cassette should be significantly higher in the hpEIN2[G:U] plants in comparison with those in the hpEIN2[WT] plants, which allow both the higher accumulation of siRNA and of large-sized dsRNA intermediates by the lowered Dicer processing.

Line 321-323: The authors mentioned that miR168 showed smaller gel mobility shift than tasiR255. The band intensities of CIP-treated tasiR255 were greatly reduced even though the band intensities of CIP-treated miR168 were nearly the same as those of non-treated miR168. Why such differences were created? In addition, comparison of the degree of mobility shift between the miR168 and tasiR255 was apparently difficult for us because of these two sRNA species were shown in different photo images.

Fig. 7E, and Table 2: What is "ocp"? The authors should explain the data of the "ocp" background plants, and should also mention in the text (ocp may be ago4-2?)

Reviewer #2:

Remarks to the Author:

The authors have improved the manuscript and addressed most of the criticisms. However, parts of the manuscript still need some work.

The northern in the new Figure 7e is not of sufficient quality and needs to be repeated – it needs a clear hpEIN2[WT] antisense siRNA showing the difference in migration rate from the hpEIN2[G:U] antisense siRNA and its size relative to the RNA markers, which are not clearly described in the legend; need to explain clearly what the first two (5'p-SM21nt and 5'p-SM24nt) and last three lanes (5'p-24nt, 5'p-24nt, 5'p-24nt) are, and how they were produced or purchased commercially; this information also needs to be clearly described in the Methods.

The plant lines in the bottom table of Figure S10 are inadequately described in the table and in the legend.

Figure 8 and Table 2 still has ocp11 instead of ago4-2. Change ocp11 to ago4-2 throughout the manuscript.

In Figure 7C legend, briefly explain what β-elimination does to change the migration of the sRNA: i.e., removes the terminal ribose only if it has both a 2' OH and a 3' OH group.

Lines 314-318: To make the following statement, the authors would need to show that the sense siRNAs derived from the hpEIN2[G:U] don't move faster on a gel than the sense siRNAs derived from the hpEIN2[WT]: "The under representation of hpEIN2[G:U]-derived antisense sRNAs compared to hpEIN2[WT]-derived antisense siRNAs in the sRNA-seq data, despite the similar or even stronger northern blot bands of hpEIN2[G:U] siRNAs (Figure 8, Figure S10, Figure S11), also suggested different 5' phosphorylation that affected 5' adaptor ligation during sRNA sequencing."

Line 349-351: Add Daxinger et al. (2007, Trends in Plant Sci 13, 4-6) reference for potential transcriptional down-regulation of 35S promoters in T-DNA insertion mutants.

Line 353: suggest changing "such" to "substantial"; also need to add the loading control for Figure 7b to the Figure 7 legend.

There are also a few typos in the new text added to the manuscript, e.g., nrpd1a-2 rather than nrpd1a-3; phosphotase instead of phosphatase.

Reviewer #3:

Remarks to the Author:

this is a second review of this manuscript following revision. The manuscript provides sufficient and compelling data of importance to the plant community with regard to RNAi technology, and the work will likely serve to enhance the efficacy of RNAi gene suppression. For these reasons, I believe that the report is suitable for publication. The changes made have enhanced clarity and, more importantly, accuracy of the report. I find the text to be improved and more careful in what it describes. I have no further suggestions and accept the responses given to my previous comments.

Response to reviewers' comments

Reviewer #1 (Remarks to the Author):

This revised manuscript was much improved. However, several points still remain to be refined.

Thank you for the positive review and the helpful suggestions.

Line 160-161: I cannot understand the differences between the following two phenomena; i.e., reduction in self silencing or increase of transcriptional stability.

Response: We modified the sentence to the following: “The enhanced uniformity of RNAi by the mismatched hpRNA transgenes suggested a reduced level of self-silencing, which could be due to reduced DNA methylation at the mismatched transgenes compared to the traditional hpRNA transgenes.”

Line 169: I do not like the “direct link between promoter methylation and reduced RNAi”. As authors showed, 5 lines (#2,5,8,9,10) showed reduced PCR amplification after McrBC treatment. However, #6 line showed similar reduction in PCR amplification when compared with the #5 line's data. The #6 line showed strong RNAi phenotype, even though this line showed moderate methylation as shown in Fig. 4C,4E. Overall, the degree of DNA methylation and reduction of RNAi show a good correlation as mentioned by authors, but I wonder if the degree of RNAi is determined only by the DNA methylation in this case.

Response: we deleted “indicating a direct link between promoter methylation and reduced RNAi”.

Line 398-400: It is likely that the large-sized dsRNA intermediates would be explained by poor Dicer processing even though the authors denied such possibility. I think that in the hpEIN2[WT] plants the transcription of inverted repeat sequences itself was suppressed by DNA methylation as indicated in this paper. So, the transcription rates of introduced inverted repeat cassette should be significantly higher in the hpEIN2[G:U] plants in comparison with those in the hpEIN2[WT] plants, which allow both the higher accumulation of siRNA and of large-sized dsRNA intermediates by the lowered Dicer processing.

Response: We agree that the possibility exists that G:U hpRNA is transcribed at high levels that could saturate Dicers leading to accumulation of unprocessed long dsRNA, but the comparable or higher siRNA abundance makes it more likely that different processing is the cause of this. We modified the statements to the following “The hpEIN2[G:U] lines accumulated high amounts of distinct large-sized dsRNA intermediates, which were largely absent in the hpEIN2[WT] lines. Furthermore, while the hpEIN2[G:U] plants accumulated

similar amounts of siRNAs to the hpEIN2[WT] lines (Figure 7B), siRNAs from the two hpRNA designs showed different gel mobilities.”

Line 321-323: The authors mentioned that miR168 showed smaller gel mobility shift than tasiR255. The band intensities of CIP-treated tasiR255 were greatly reduced even though the band intensities of CIP-treated miR168 were nearly the same as those of non-treated miR168. Why such differences were created? In addition, comparison of the degree of mobility shift between the miR168 and tasiR255 was apparently difficult for us because of these two sRNA species were shown in different photo images.

Response: Thank you for the comments. We modified Figure 7D by adding two short red lines indicating the average positions of the CIP-treated and untreated sRNA bands, and modified the figure legend by adding “The two short red lines on the right indicate the average position of the CIP-treated and untreated sRNAs, which show a bigger gap or gel shift for the tasiR255 bands than for the miR168 bands.” We noticed the strongly reduced band intensity of the tasiR255 band in the CIP-treated samples that was different to the miR168 band, but could not provide a good explanation for it. We added this to the figure legend: “It was unclear why the intensity of tasiR255 band was strongly reduced upon CIP treatment.”

Fig. 7E, and Table 2: What is “ocp”? The authors should explain the data of the “ocp” background plants, and should also mention in the text (ocp may be ago4-2?)

Response: We changed “ocp” to “ago4-2” as suggested by Reviewer 2, but missed these ones in the last revision – changes are in this new version.

Reviewer #2 (Remarks to the Author):

The authors have improved the manuscript and addressed most of the criticisms. However, parts of the manuscript still need some work.

Thank you again for the positive review, and for the helpful suggestions.

The northern in the new Figure 7e is not of sufficient quality and needs to be repeated – it needs a clear hpEIN2[WT] antisense siRNA showing the difference in migration rate from the hpEIN2[G:U] antisense siRNA and its size relative to the RNA markers, which are not clearly described in the legend; need to explain clearly what the first two (5’p-SM21nt and 5’p-SM24nt) and last three lanes (5’p-24nt, 5’p-24nt, 5’p-24nt) are, and how they were produced or purchased commercially; this information also needs to be clearly described in the Methods.

Response: Thank you for the suggestions. We performed additional northern blot hybridization, which is presented in the revised Figure 7e and Supplementary Figure 11b. In the new northern blot, we included both unphosphorylated and phosphorylated sRNA

markers. We also added total RNA from WT Col-0 to all the sRNA marker samples to ensure even running of all samples in the gel. The result showed clearly that 5' phosphate increases the gel mobility of sRNAs so the di- and tri-phosphorylated sRNA markers migrated faster than the mono- and un-phosphorylated markers. The shift in gel mobility between hp[WT] and hp[G:U] derived siRNAs also became clearer in the new northern blots of Figure 7e and Supplementary Figure 11b. Figure legends have been modified to include more information on the β -elimination assay and sRNA markers.

The plant lines in the bottom table of Figure S10 are inadequately described in the table and in the legend.

Response: We revised the legends to include more information about the plant lines. The numbering of the hpEIN2 lines of the Col-0 background in the previous version of Figure S10 was based on the numbers of RNA preparations rather than line numbers shown on Figure 2, which have now been corrected.

Figure 8 and Table 2 still has ocp11 instead of ago4-2. Change ocp11 to ago4-2 throughout the manuscript.

Response: We have now changed all “ocp11” or “ocp” to “ago4-2”.

In Figure 7C legend, briefly explain what β -elimination does to change the migration of the sRNA: i.e., removes the terminal ribose only if it has both a 2' OH and a 3' OH group.

Response: We added the info in Figure 7c legends.

Lines 314-318: To make the following statement, the authors would need to show that the sense siRNAs derived from the hpEIN2[G:U] don't move faster on a gel than the sense siRNAs derived from the hpEIN2[WT]: “The under representation of hpEIN2[G:U]-derived antisense sRNAs compared to hpEIN2[WT]-derived antisense siRNAs in the sRNA-seq data, despite the similar or even stronger northern blot bands of hpEIN2[G:U] siRNAs (Figure 8, Figure S10, Figure S11), also suggested different 5' phosphorylation that affected 5' adaptor ligation during sRNA sequencing.”

Response: To avoid confusion, we deleted this statement and the associated early version of Figure S11.

Line 349-351: Add Daxinger et al. (2007, Trends in Plant Sci 13, 4-6) reference for potential transcriptional down-regulation of 35S promoters in T-DNA insertion mutants.

Response: We added this reference.

Line 353: suggest changing “such” to “substantial”; also need to add the loading control for Figure 7b to the Figure 7 legend.

Response: Change was made accordingly.

There are also a few typos in the new text added to the manuscript, e.g., nrpd1a-2 rather than nrpd1a-3; phosphotase instead of phosphatase.

Response: Corrections were made.

Reviewer #3 (Remarks to the Author):

this is a second review of this manuscript following revision. The manuscript provides sufficient and compelling data of importance to the plant community with regard to RNAi technology, and the work will likely serve to enhance the efficacy of RNAi gene suppression. For these reasons, I believe that the report is suitable for publication. The changes made have enhanced clarity and, more importantly, accuracy of the report. I find the text to be improved and more careful in what it describes. I have no further suggestions and accept the responses given to my previous comments.

We very much appreciate the positive review.

Reviewers' Comments:

Reviewer #1:

Remarks to the Author:

Now, I believe that this manuscript is acceptable for publication.

Reviewer #2:

Remarks to the Author:

The authors have addressed all of my criticisms and recommend that the manuscript be accepted for publication.